# STEER-ME: Assessing the Microeconomic Reasoning of Large Language Models

**Narun K. Raman, Taylor Lundy, Kevin Leyton-Brown**
Department of Computer Science
University of British Columbia
Vancouver, BC, Canada
{narunram,tlundy,klb}@cs.ubc.ca

**Jesse Perla**
Department of Economics
University of British Columbia
Vancouver, BC, Canada
jesse.perla@ubc.ca

## Abstract

Large language models (LLMs) are increasingly being asked to make economically rational decisions and indeed are already being applied to economic tasks like stock picking and financial analysis. Existing LLM benchmarks tend to focus on specific applications, making them insufficient for characterizing economic reasoning more broadly. In previous work, we offered a blueprint for comprehensively benchmarking *strategic* decision-making Raman et al. [2024]. However, this work did not engage with the even larger microeconomic literature on *non-strategic* settings. We address this gap here, taxonomizing microeconomic reasoning into 58 distinct elements, each grounded in up to 10 distinct domains, 5 perspectives, and 3 types. The generation of benchmark data across this combinatorial space is powered by a novel LLM-assisted data generation protocol that we dub `auto-STEER`, which generates a set of questions by adapting handwritten templates to target new domains and perspectives. By generating fresh questions for each element, `auto-STEER` induces diversity which could help to reduce the risk of data contamination. We use this benchmark to evaluate 27 LLMs spanning a range of scales and adaptation strategies, comparing performance across multiple formats—multiple-choice and free-text question answering—and scoring schemes. Our results surface systematic limitations in current LLMs' ability to generalize economic reasoning across types, formats, and textual perturbations, and establish a foundation for evaluating and improving economic competence in foundation models.

## 1 Introduction

There has been much recent interest in both assessing and improving the decision-making abilities of LLMs. Over the past century or so, the quantitative study of optimal decision-making has largely been conducted in the field of economics. This study has yielded a wide range of normative principles, identification of qualitatively different settings in which different such principles apply, and robust senses in which human decision-makers systematically tend to fall short of the optimal ideal. It is thus both natural and important to assess the decision-making performance of LLMs according to the yardstick of economic rationality. Beyond this abstract sense in which economics is important to LLMs, there additionally exist a wide range of economic application areas in which LLMs are already being fruitfully applied. Some prominent examples include financial sentiment analysis, where LLMs are tasked with analyzing the sentiment information of financial texts [Malo et al., 2013, Maia et al., 2018, Araci, 2019, Yang et al., 2020]; question answering, where LLMs are tasked with answering an economic question based on the provided information [Maia et al., 2018, Chen et al., 2021, 2022, Shah et al., 2022, Xie et al., 2023b, Raman et al., 2024]; financial text summarization, which entails condensing long unstructured financial texts into short summaries that capture crucial information and maintain factual consistency with the original long texts [Mukherjee et al., 2022,

Zhou et al., 2021]; and Named Entity Recognition, which asks the model to detect critical financial entities such as persons, organizations, and locations [Alvarado et al., 2015, Shah et al., 2022]. More open-ended applications are also starting to emerge. LLMs such as WallStreetBERT, TradingGPT, FinGPT, FinTral, and BloombergGPT are already giving advice to investors and financial advisors [Xie et al., 2023a, Li et al., 2023, Yang et al., 2023, Bhatia et al., 2024, Wu et al., 2023]. LLMs are also being deployed as agents to automate budgetary planning and allocation [Chen et al., 2023] as well as in simulations to analyze the impact of policy changes on key indicators like inflation and GDP growth [Carriero et al., 2024, Li et al., 2024a].

It is thus important to be able to determine when we should trust an LLM in such an open-ended economic application, or more broadly when we should consider that an LLM meets the standard of economic rationality. We argue that a necessary condition is that it should demonstrate robust performance on the fundamentals of economic reasoning, just as, e.g., human experts in finance and economics are required to do. Many existing benchmarks have been proposed, many of which were introduced in papers cited above, but most focus narrowly on a single task or application rather than assessing economic reasoning more broadly. A second—useful but insufficient—category of benchmarks tests foundational concepts in mathematics, ranging from basic arithmetic to complex problem-solving [Huang et al., 2016, Ling et al., 2017, Amini et al., 2019, Lample and Charton, 2019, Zhao et al., 2020]. Notable examples include GSM8K [Cobbe et al., 2021], a small but varied dataset of moderately difficult math problems, and MATH [Hendrycks et al., 2021], a challenging benchmark at the time, where no evaluated model had reached expert-level performance. Recent state-of-the-art models routinely achieve pass@$k$ accuracies exceeding human performance on these benchmarks. However, such pass@$k$ metrics are inherently optimistic—they credit models if any of $k$ sampled completions is correct, yielding only an upper bound on true capability. Metrics that reflect only best-case performance can mislead users about model robustness and provide an untrustworthy basis for deploying LLMs in high-stakes economic applications. Trustworthy deployment demands more than narrow competence; it requires comprehensive evaluations.

What might comprehensive assessment of an LLM's economic reasoning look like? We propose three guiding principles. First, evaluations should include diverse economic domains. Economics involves problems such as determining optimal consumption, forecasting profits under uncertainty, and analyzing supply shifts on equilibrium prices. Moreover, LLM performance can vary significantly across closely related tasks or subtle textual changes [e.g., Hendrycks et al., 2020a, Ribeiro et al., 2020]. For instance, LLMs proficient in maximizing profit might falter at minimizing cost, or perform inconsistently across similar budget-allocation tasks when contexts differ. Second, evaluations should span different LLM adaptation strategies and metrics to verify that the performance metrics reflect true behavior rather than artifacts of particular adaptations or scoring schemes [Ethayarajh and Jurafsky, 2020, Lester et al., 2021, Islam et al., 2023, Schaeffer et al., 2023, Bhatia et al., 2024]. In a setting where precise quantitative reasoning is critical, differences in adaptation—such as whether a model is permitted to call external tools for computation or perform chain-of-thought reasoning—can substantially alter performance. Finally, assessments must be robust to variations in testing format and environment, guarding against biases introduced by specific paradigms. This is especially important in economics, where obtaining correct answers often requires multi-step reasoning and where validating correctness is often easier than producing a correct solution. For example, comparing multiple-choice question answering (MCQA) versus free-text question answering (FTQA) can reveal whether high scores arise from superficial cues or genuine economic reasoning. By organizing benchmarks around these parallel desiderata, we can build assessments that yield reliable measures of economic reasoning.

Our own recent work [Raman et al., 2024] developed a benchmark distribution—dubbed STEER—for assessing economic reasoning in strategic settings, which aimed for comprehensiveness in the first two principles. This work began by taxonomizing the fundamental concepts underlying game theory and decision theory into 64 distinct "elements of economic rationality," ensuring that the elements in the benchmark covered a wide range of strategic contexts and decision-making problems. It furthermore formalized a hierarchy across elements so that an LLM's performance could be better understood in the context of its dependent subtasks. We generated a huge set of questions from this taxonomy, which vary in their difficulty and domain (e.g., finance, medicine, public policy). Crucially, to assess reasoning robustness rather than one-off success, we queried each question set with hundreds of semantic and numeric perturbations, requiring models to consistently derive the correct answer across varied forms. The paper concluded with an evaluation of 28 LLMs over two

adaptation strategies and scored with a suite of metrics. We defined this evaluation framework as a `STEER Report Card` (`SRC`), a flexible scoring rubric that can be tuned by the user for their particular needs.

While `STEER` is comprehensive in its taxonomy, hierarchy, and robustness checks, it has a key limitation: it neglects much of the subject matter of microeconomics. This blind spot is multiagent settings in which agents nevertheless act nonstrategically. Such reasoning is widespread in competitive markets, where each agent's impact on the market is too small to affect prices unilaterally. For example, while a mobile phone manufacturer might make a strategic decision about the number of handsets to produce and the price to sell them at, a small farm's decision to produce wheat instead of corn given market prices is non-strategic. To address these gaps, this paper employs and extends the `STEER` blueprint to construct a benchmark for non-strategic economic reasoning. Specifically, we built a taxonomy of non-strategic economics consisting of $58$ elements. We then instantiated each element in the taxonomy across 8–10 domains. From here, we expanded on the blueprint in two ways. First, we increased the diversity of the questions in the dataset and instantiated each element in $5$ different *perspectives* and up to $3$ *types* (as defined in Section 3.1). Second, we expanded STEER's evaluation framework by encoding every question in both MCQA and FTQA format, included newer LLMs ($27$ in total) and new scoring metrics (a family of $4$ calibration metrics).We dub our benchmark `STEER-ME`, reflecting both its conceptual links to the original `STEER` and its novel focus on microeconomics.

There are two other conceptual senses in which the current paper goes beyond our previous work on STEER. First, even given the best possible LLM benchmark, data contamination poses an increasingly important challenge [Sainz et al., 2023, Deng et al., 2023, Ravaut et al., 2024]. Data contamination occurs when the test data used to evaluate an LLM is similar or identical to data the LLM encountered during training, leading to inflated performance metrics that do not accurately reflect the LLM's true capabilities. To tackle this issue, we introduce a new dynamic data generation process called `auto-STEER` which we used to generate all of the questions in `STEER-ME`. `auto-STEER` combines many of the features present in existing dynamic and modular frameworks [Gioacchini et al., 2024, Wang et al., 2024, White et al., 2024] that we detail in Appendix B. Second, STEER relies exclusively on MCQA, offering no ability to explore biases arising from the form in which questions are posed. As we will see, this distinction matters profoundly for evaluating a model's economic reasoning.

In what follows, Section 2 gives an overview of our taxonomy; for space reasons we defer definitions and examples of each element—which are extensive—to Appendix A. Section 3 describes how we used this taxonomy to build the benchmark distribution. For 37 elements, we have written LLM prompts to synthetically generate $1,000$–$5,000$ multiple-choice questions and manually validated $500$ generations per element. Section 4 describes the setup of an experiment in which we generated full `SRC`s for 28 LLMs, ranging from Llama-2 7B to o1-preview, evaluated on a total of $21,000$ test questions. We spent $\$12,439.54$ making requests to OpenAI and Anthropic's API and 9.81 GPU years of compute to evaluate open-source models. Finally, we survey our experimental results in Section 5. Here, we offer a few highlights: (i) Even though an LLM may look excellent when you grant it $k$ tries per question, its actual reasoning lacks robustness: changes in the domain or in parameter values often cause performance loss; (ii) MCQA questions are susceptible to exploitation, leading to inflated performance as LLMs rely on shortcuts such as substituting provided options; (iii) FTQA questions, by removing such shortcuts, can serve as an effective diagnostic tool to pinpoint specific conceptual misunderstandings and reasoning errors (e.g., defaulting to Marshallian rather than Hicksian demand derivations); and (iv) performance on foundational mathematical tasks—such as exponent manipulation—strongly predicts accuracy on more complex economic reasoning problems within our benchmark.

We will release all model outputs to support evaluation research and contributions via our website `steer-benchmark.cs.ubc.ca`, allowing users to deeply probe all of our experimental results and the underlying model prediction details. Finally, we will release an extensible codebase to support the community in taking `STEER-ME` further.

## 2   Elements of Microeconomic Reasoning

Our first step in generating a benchmark for non-strategic microeconomics is to taxonomize this space. Previous work by Raman et al. [2024] developed a taxonomy for economic rationality within strategic domains. Their approach involved identifying foundational principles that define how agents should

| **Setting 1: Foundations** | |
|---|---|
| **Module 1.1:** Optimization ❶ | Number of elements: 6 |
| **Module 1.2:** Systems of Equations ❶ | Number of questions: $127,342$ |
| **Module 1.3:** Derivatives and Homotheticity ❶ | Number of types: 1 |
| **Setting 2: Consumption Decisions in Non-Strategic Environments** | |
| **Module 2.1:** Properties of Utility Functions ❶ | Number of elements: 22 |
| **Module 2.2:** Deriving Demand ❶ | # of questions: $3,295,770$ |
| **Module 2.3:** Comparative Statics of Demand ❶ | Number of types: 14 |
| **Module 2.4:** Labor Supply ❶ | |
| **Module 2.5:** Dynamic Consumption Decisions ❶ | |
| **Setting 3: Production Decisions in Non-Strategic Environments** | |
| **Module 3.1:** Properties of Production Functions ❶ | Number of elements: 16 |
| **Module 3.2:** Deriving Factor Demand ❶ | # of questions: $1,333,330$ |
| **Module 3.3:** Comparative Statics with Production ❶ | Number of types: 20 |
| **Module 3.4:** Dynamic Production Decisions ❶ | |
| **Setting 4: Non-Strategic Decisions in Multi-Agent Environments** | |
| **Module 4.1:** Consumer Goods Market Aggregation ❶ | Number of elements: 10 |
| **Module 4.2:** Factor Market Aggregation ❶ | # of questions: $750,060$ |
| **Module 4.3:** Prices in Static Market Equilibrium ❶ | Number of types: 6 |
| **Module 4.4:** Comparative Statics of Equilibrium Prices ❶ | |
| **Setting 5: Evaluating Equilibria and Externalities** | |
| **Module 5.1:** Welfare and Decentralization ❶ | Number of elements: 10 |
| **Module 5.2:** Welfare Analysis of Market Equilibrium ❶ | # of questions: $698,367$ |
| | Number of types: 5 |

Table 1: High-level diagram of the taxonomy of elements of microeconomic reasoning. At the top level, we divide the space of decision making into 5 settings; we further subdivide settings into modules (e.g., Comparative Statics of Demand) that capture conceptually similar behaviors. We also include a few summary statistics about the dataset. Each ❶ icon is a hyperlink to the corresponding module in our appendix.

make decisions in specific environments and then organizing these principles, or "elements," into progressively more complex decision-making scenarios. We adopt a similar hierarchical approach for STEER-ME, focusing on organizing economic decision-making principles into structured categories. However, unlike STEER, which assesses decision-making in strategic environments, our focus is assessing how agents make decisions given prices and quantities that are determined by the forces of supply and demand. We call this sub-field non-strategic microeconomics.

Two of the settings from STEER remain directly relevant to non-strategic microeconomics: FOUNDATIONS and DECISIONS IN SINGLE-AGENT ENVIRONMENTS. As we describe our taxonomy, we begin with these foundational settings. The elements we incorporate from FOUNDATIONS—arithmetic, optimization, probability, and logic—are core mathematical skills essential for microeconomic reasoning and are already present in STEER. In STEER-ME, we expand this setting by adding elements that test basic calculus, such as single-variable derivatives and linear systems of equations. In STEER, DECISIONS IN SINGLE-AGENT ENVIRONMENTS focused on testing whether an agent can adhere to the von Neumann-Morgenstern utility axioms when making decisions over a set of alternative choices. We include those axiomatic elements and extend this setting to include testing the properties of commonly used parameterizations of utility functions in non-strategic microeconomic contexts, such as utility functions with satiation points, monotone preferences, and budget constraints.

Building directly on these foundational settings, we introduce the next setting, DECISIONS ON CONSUMPTION IN NON-STRATEGIC ENVIRONMENTS, which tests an agent's ability to optimally exchange time and money for desired goods and services. Elements in this setting assume that the agent is a price taker, meaning that the agent accepts market prices as given rather than forecasting how a purchase might move the market. First, we test the agent's ability to derive demand functions consistent with the axioms and functional forms from DECISIONS IN SINGLE-AGENT ENVIRONMENTS. These foundational elements are useful in assessing whether an agent can make consistent, rational choices in response to market prices. We then include elements testing the agent's ability to determine optimal consumption bundles, decide when to leave the workforce, and conduct comparative statics with demand functions.

DECISIONS ON PRODUCTION IN NON-STRATEGIC ENVIRONMENTS tests an agent's ability to decide on the combination of inputs to efficiently produce goods and services to maximize profits. The setting starts by assessing the agent's ability to identify and analyze basic properties of production functions, such as the relationship between input quantities and output levels. This includes concepts like returns to scale, diminishing marginal returns, and the technological constraints that shape production capabilities. We then test the agent's ability to conduct expenditure minimization and its dual, profit maximization. This involves solving optimization problems where the agent must use marginal analysis to determine the quantity of output that maximizes profit (i.e., minimizes cost).

DECISIONS IN MULTI-AGENT NON-STRATEGIC ENVIRONMENTS considers consumers and producers who each reason according to the principles just described to trade with each other. This more complex setting requires an agent to reason about how the aggregated behaviors of consumers and producers lead to market-clearing prices that balance supply and demand. This setting covers elements such as finding market-clearing prices, computing competitive equilibria, and analyzing the comparative statics of equilibrium in markets where individual actions do not directly impact others.

Our last setting, EVALUATING EQUILIBRIA AND EXTERNALITIES, tests agents on their ability to evaluate whether equilibria are efficient and to analyze the effects of interventions, such as taxes or price ceilings, on welfare. In this setting, agents must not only be able to analyze how supply and demand dynamics establish equilibrium prices but also consider how external interventions shift these dynamics and alter the behavior of both consumers and producers. The elements in this setting can be relatively simple (e.g., compute consumer/producer surplus) or involve detailed counterfactual analysis (e.g., predict how interventions impact prices, the allocation of resources, and welfare outcomes).

For a more detailed discussion on the structure of these elements and the methodology we used to group the elements, including formal definitions, we refer the reader to Appendix A.

## 3   The STEER-ME Benchmark

We first give an overview of STEER-ME dataset and then explain the process we used to generate and validate these questions, which we call auto-STEER. Finally, we describe our evaluation framework.

### 3.1   Dataset

We implement our benchmark in both Multiple-Choice Question Answering (MCQA) and Free-Text Question Answering (FTQA) formats, enabling direct comparisons across evaluation protocols. The MCQA format aligns with widely used conventions in prior work [see, e.g., Rajpurkar, 2016, Wang et al., 2018, 2019, Zellers et al., 2019, Hendrycks et al., 2020b, Shah et al., 2022, Liang et al., 2022, Suzgun et al., 2022], while the FTQA format follows practices established in recent benchmarks [see, e.g., Nan et al., 2022, Reddy et al., 2019, Choi et al., 2018]. In MCQA, each test question presents a decision-making scenario along with several candidate options, where only one is correct. One key advantage of MCQA is its simplicity: models return a single token corresponding to an answer choice, which makes outputs easy to parse and allows metrics based on token probabilities, such as expected calibrated error, to be more meaningful and interpretable [Liang et al., 2022, Li et al., 2024b]. Thus, MCQA serves as a useful baseline that contextualizes our findings within the broader literature. The FTQA setting, by contrast, requires the model to produce answers without relying on pre-specified choices. Although FTQA introduces potential challenges in answer grading, our benchmark is particularly amenable to FTQA because answers are typically numeric, allowing us to apply automatic, rule-based grading functions (e.g., exact numeric match or symbolic equivalence). Moreover, FTQA provides an important complementary perspective, as it assesses model reasoning without the structural cues embedded in multiple-choice options. By including both formats, we gain deeper insights into model capabilities and robustness, balancing comparability with prior studies (via MCQA) against a stricter evaluation of reasoning processes (via FTQA).

Furthermore, diagnosing where and why a LLM fails in standard MCQA formats is inherently challenging because the LLM's response is typically just an option letter, providing limited insight into the underlying reasoning or the actual computed answer. Although reasoning trajectories could potentially be analyzed to extract detailed answers, the mere presence of multiple-choice options can bias models in unpredictable ways. We demonstrate evidence of this phenomenon in Section 5.1.

**Question:**

Sophie is buying textbooks for her university classes, her demand for textbooks at any given price is expressed by the following demand function {d_function}. What is Sophie's consumer surplus if the price of textbooks is {price}?

*Domain: Education, Perspective: Third Person Woman*

**Question:**

John is purchasing hockey sticks, his demand for hockey sticks at any given price is expressed by the following demand function {d_function}. What is John's consumer surplus if the price of hockey sticks is {price}?

*Domain: Sports, Perspective: Third Person Man*

Figure 1: This figure depicts two questions in the consumer surplus element with different domains and perspectives. The text colored in red are the labeled fields that will be filled for test time and the text in blue is the perspective. On top, a question is framed in the education domain from a third-person woman perspective, while on the bottom, the same question is written for the sports domain from a third person man perspective. These were both generated during the style-transfer step in the data generation process.

FTQA, on the other hand, naturally facilitates error classification and quantification since models explicitly produce their reasoning paths without being influenced by predetermined options. This transparency enables researchers to accurately pinpoint specific reasoning mistakes and quantify their occurrence, which is considerably more difficult with traditional MCQA.

Our own benchmark consists of a total of 37 instantiated elements, each containing 5,000–20,000 questions. Each question is characterized by a (type, domain, perspective) tuple. Different *types* represent distinct ways of testing an agent's abilities within an element. For example, we could assess an agent's ability to perform profit maximization by asking "What is the maximum profit?" or "How much labor is needed to maximize profit?" Therefore, an (element, type) tuple represents a distinct concept being tested. The *domain* of a question indicates which of 10 predefined topic areas it pertains to: consumer goods, medical, finance, education, technology, entertainment, environmental policy, politics, sports, or gambling. Finally, the *perspective* of a question represents which of the 5 predefined perspective the question was written in: first-person, second-person, third-person anonymous, third-person female and third-person male. We disallow over (type, domain, perspective) combinations that do not lead to coherent questions; for example, questions about welfare theorems do not make sense in gambling settings.

## 3.2 `auto-STEER`

Similar to our generation process in `STEER` Raman et al. [2024], we leveraged a state-of-the-art LLM to help generate our dataset. We substantially extended their methodology, however, by adding an additional style-transfer step where we asked the LLM to rewrite questions in new domains or perspectives. This greatly increased the variety of questions we were able to add. This section describes how we used our new approach to design `STEER-ME`.

First, for each type we hand-wrote a set of gold-standard example templates that served as the seeds for the data generating process. As can be seen in Figure 10, these templates were tagged with a domain, a perspective, and a type, if appropriate. The majority of these questions had *labeled fields* for numbers (e.g., "...the cost of labor is {cost}...") which were programmatically filled for test time. See Figure 1 for an example.

Next, we asked the LLM to style-transfer these templates into each of the domains. We primarily leveraged gpt-4o to generate our benchmark, but as we show in Appendix I.1, using claude-3-5-sonnet did not change LLM performance. Our prompt included explicit instructions to maintain the same set of labeled fields as the hand-written templates. Figure 2 depicts the style-transfer page in our web application along with the prompting instructions. LLMs can be inconsistent in maintaining the economic meaning of questions after domain style transfer, so we hand-checked each of the outputted templates and edited them when necessary. All of these operations are supported by a web application we built: see figures in Appendix K. We then further style-transferred each of these newly generated templates into each perspective, resulting in up to 40 unique domain-perspective pairs for each type. We ran an additional check on the style-transfer process by filling the labeled fields in the templates with values and asking the LLM to solve the questions as written, which we found could highlight mistakes in question wording or in programmatically filled values; see Figure 11 in Appendix K.(We were careful only to use his procedure to correct mistakes in the templates, not to tune the difficulty of the questions in a way that would bias our benchmark.)

We then took each of these templates and asked the LLM to replicate the template, keeping the domain, perspective and labeled fields fixed but modifying exact words or objects used in the question. We generated 100 new templates for each element, crossing every domain and perspective pair, resulting in 30,000 templates across the dataset. We then spot-checked 500 of the resulting templates for each element, and flagged 99.88 % of the templates as valid. Applying a simple Chernoff bound, we estimate with 95 % confidence that between 97.1 % and 99.4 % of all generated questions are valid.

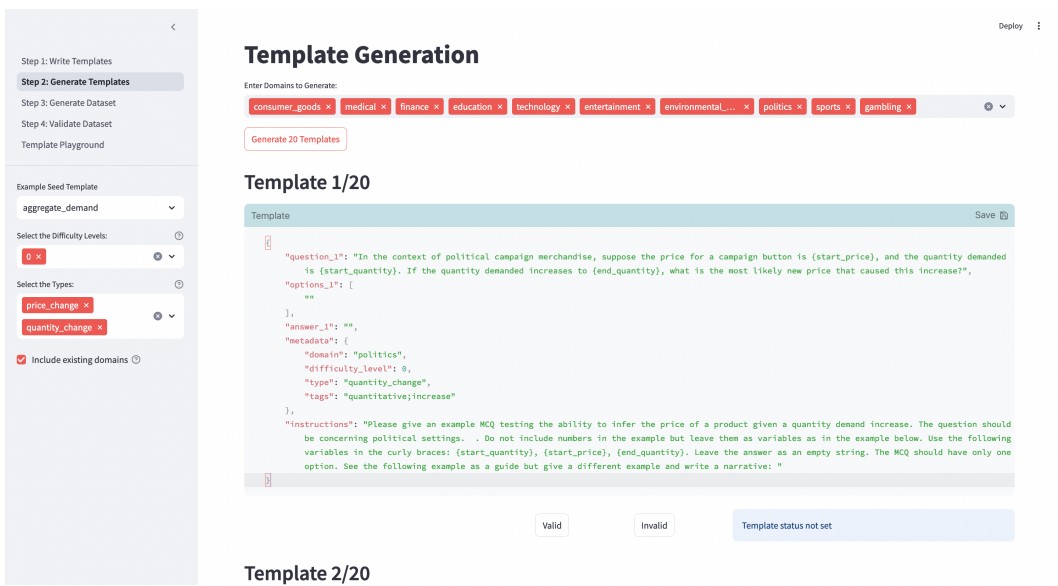

Figure 2: The web app user interface for template generation. This page allows for a selection of domains, and types for which templates will be generated using the available example seeds. Templates can then be verified and saved by the user.

Finally, we created 20 instantiated questions from each template by filling its labeled fields with randomly generated values. We restricted the random generator to output numbers that were appropriate given the context: e.g., demand functions had negative slopes, positive values for equilibrium prices, etc. We programmatically solved each question and filled in the appropriate options and answer. In the end, we produced 1,000 questions per (domain, perspective) pair and up to 40,000 per type.

### 3.3   Scoring

Given a complete set of model responses, it is far from straightforward to choose a way of computing a single, overall performance score. Consequently, benchmarks often employ a suite of metrics to provide a more comprehensive assessment of performance [Wang et al., 2019, Gehrmann et al., 2021, Liang et al., 2022, Srivastava et al., 2022]. We include a longer discussion as well as definitions of our scoring metrics in Appendix C and simply list the metrics here: Exact-match accuracy, Normalized accuracy, Expected calibration error, Brier score, and Expected probability assignment.

A LLM's score on an element is the average taken over all questions in an element. We consider an element a base concept in our benchmark and therefore define the accuracy and confidence metrics with respect to an element.

## 4   Experimental Setup

Table 7 in Appendix H lists the 28 LLMs we evaluated. We ran gpt-4o, gpt-4o-mini, o1-preview, and o3 using OpenAI's API [OpenAI, 2020]; claude-3-5-sonnet and claude-3-haiku using Anthropic's API [Anthropic, 2025]. We obtained 22 open-source LLMs from the HuggingFace Hub [Wolf et al., 2019] and ran them on between 1 and 4 L40 GPUs. Due to time and budget constraints we evaluated the closed-source reasoning LLMs (e.g., o1-preview, o3) on 13 elements and the remaining models on all of the instantiated elements.

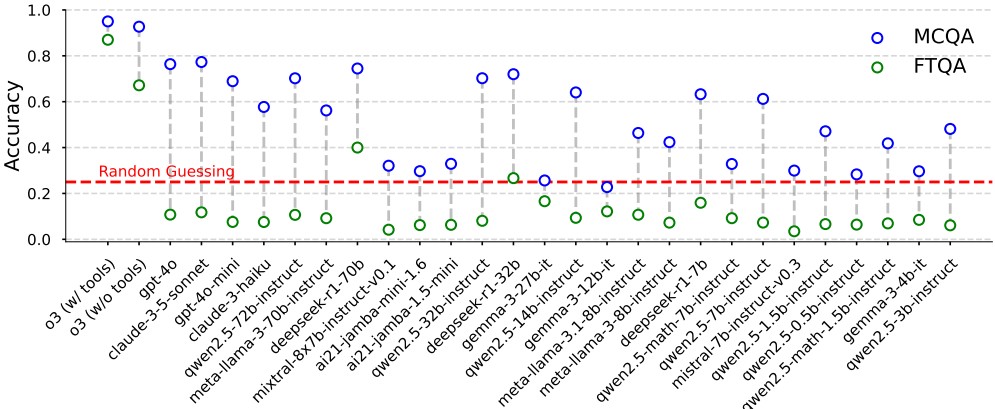

Figure 3: This figure plots the gap in accuracy between MCQA and FTQA for all models across our benchmark. We use the exact-match accuracy metric for both MCQA and FTQA and plot the accuracy random guessing would have gotten on MCQA in red. We sort models on the x-axis by parameter count when we have it and alphabetically otherwise.

## 5 Results

Figure 3 summarizes model performance on MCQA and FTQA across all tested elements. We highlight three trends. First, reasoning-tuned models (e.g., o3, o1-preview, DeepSeek R1) substantially outperform standard LLMs on both MCQA and FTQA—even though o3 and o1-preview are evaluated on a harder subset, they remain the top performers. Second, computational tools disproportionately aid FTQA: removing tools lowers o3's FTQA accuracy by $15.2$ percentage points versus only $1.7$ percentage points on MCQA. Tools chiefly mitigate sensitivity to numerical parameterizations; an analysis of variance indicates that up to $91\%$ of o1-preview's performance variance is attributable to numerical instantiations, and $21\%$ of DeepSeek R1-70B's variance to different domains (Appendix G). This pattern suggests that numerics are a genuine source of brittleness, but that the core challenge in our dataset remains economic reasoning. Third, despite o3's strong results, many widely used smaller models still struggle, leaving sizable and persistent gaps. Taken together, these findings caution that, without systematic prompt rephrasing and parameter sweeps, one can easily overestimate the consistency and generalization of LLM's economic reasoning.

### 5.1 MCQA *vs.* FTQA

To disentangle true economic reasoning from option-driven shortcuts, we leverage the paired structure of our benchmark: each MCQA question has a corresponding FTQA variant that withholds choices and requires the model to provide a fully derived answer. Comparing performance across these formats, visualized in Figure 3, reveals a persistent gap, with LLMs consistently performing better on MCQA. Accounting for random guessing explains $57\%$ of the gap at an aggregate level. However, this high-level adjustment does not tell us which specific items were guessed versus those influenced by other factors. To identify the causes of the remaining discrepancy, we conducted a deeper analysis. First, we observed that LLMs select the "closest" answer option $53.38\%$ of the time. Then, focusing on the errors that persist beyond both random guessing and "closest-answer" behavior, we identified two mechanisms driving the residual performance gap: option "gaming" and contextual anchoring.

**Option gaming.** In complex elements, LLMs often failed to arrive at correct solutions through genuine optimization. Instead, we observed a "cheating" strategy in which the model, when appropriate, would plug in the candidate answer choices directly into the functions in the question and pick whichever yielded the best result, rather than solving via first principles (e.g., by taking derivatives). This was particularly evident in the Profit Maximization element, where models were asked to determine the labor input that maximizes profit. Rather than identifying the profit-maximizing choice analytically, they simply tested each option and selected the highest-profit outcome. Indeed, every spot-checked instance of a correct response from gpt-4o and claude-3-5-sonnet used this shortcut (see Appendix E.3 for an example). While this approach is a common strategy for test-takers—humans

and LLMs alike—it bypasses the economic reasoning that these elements were designed to assess. When evaluated on FTQA, no model other than o3 (with tools) exceeded $10\,\%$ exact-match accuracy.

**Contextual anchoring.** LLMs appeared to leverage multiple-choice options when reasoning about aggregating demand or supply functions. This behavior was particularly evident in the Aggregation of Consumer Demand element, which explicitly involves summing individual consumer demand functions across a large population. A common conceptual mistake that LLMs made was computing the demand for a single consumer rather than the sum—both operations frequently appear in economic reasoning contexts. However, when provided multiple-choice options, models could use numeric cues embedded in the provided options to infer the correct aggregation method. Appendix F illustrates this phenomenon quantitatively: as the number of digits in the correct sum increased from 5 to 6, claude-3-5-sonnet's exact-match accuracy rose dramatically from $38\,\%$ to $72\,\%$. This pattern indicates that larger numeric answer choices served as an implicit signal (or "nudge") to models, guiding them toward correctly performing the summation rather than mistakenly calculating an average. This finding underscores the extent to which even advanced LLMs rely heavily on textual and numeric cues, highlighting a potential vulnerability in tasks that are otherwise conceptually straightforward.

## 5.2 FTQA as a Diagnostic

Many benchmark elements proved challenging, even in MCQA format, as LLMs often simplified or misinterpreted the problem. When analyzing FTQA responses, in $36.2\,\%$ of claude-3-5-sonnet's incorrect responses to Intertemporal Consumption Smoothing, the model collapsed multi-period cash flows into a single period. Additionally, LLMs frequently ignored crucial aspects specified in the problem, such as risk preferences. Rather than appropriately considering risk aversion, the LLMs instead employed simpler assumptions, as if the decision-maker were indifferent to risk. This occurred in $11.0\,\%$ of claude-3-5-sonnet's and $42.7\,\%$ of gpt-4o's responses. See Appendix I.2.

A similar issue arose in deriving Hicksian demand functions. LLMs often reverted to the more familiar Marshallian derivation. gpt-4o used the Marshallian approach in $51.2\,\%$ of its incorrect answers, compared to $23.4\,\%$ for gpt-4o-mini. Among Anthropic models, claude-3-5-sonnet did so in $48.9\,\%$, versus $15.2\,\%$ for claude-3-haiku. These rates likely reflect varying model capacities; more capable models produced more relevant—but not always correct—responses. Note that we can only identify if an LLM as using the Marshallian approach when it came to the right 'Marshallian' answer.

Unexpectedly, none of the closed-source LLMs, except reasoning models, consistently solved Deadweight Loss of a Monopoly, a task whose primary mathematical requirement is computing the area of a triangle. Models like claude-3-5-sonnet and gpt-4o often applied incorrect formulas and misinterpreted marginal cost. Figure 4 shows that gpt-4o often compounded these errors, while claude-3-5-sonnet relied on a particular incorrect formula in nearly $44\,\%$ of responses. Full details are in Appendix I.3.

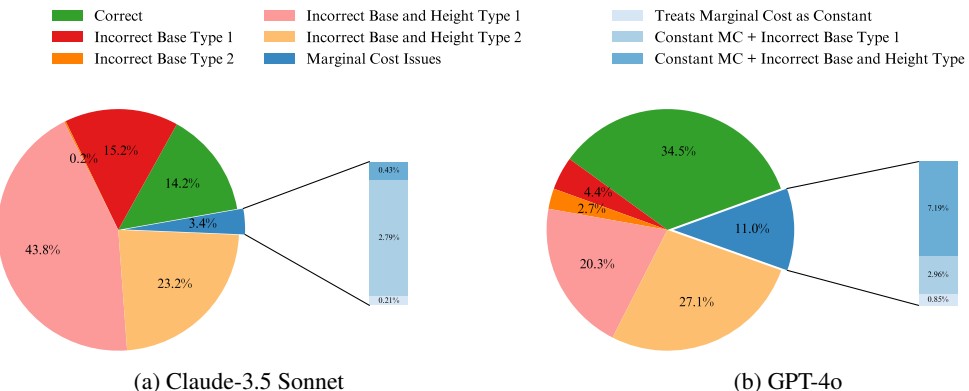

(a) Claude-3.5 Sonnet               (b) GPT-4o

Figure 4: Error analyses of claude-3-5-sonnet and gpt-4o on the Deadweight Loss of a Monopoly element. In reds and oranges are failures due to incorrect computations of the deadweight loss area; in blue and further broken down are errors due to incorrectly interpreting the marginal cost. A more detailed description of what each error means can be found in Appendix I.3.

These findings underscore a broader concern: when faced with more complex tasks, sophisticated LLMs often give "near-miss" solutions that can appear correct at first glance but actually solve subtly different problems. This is especially worrisome in real-world use cases where a user may not easily be able to verify an answer.

### 5.3 Foundational Competence Predicts Downstream Performance

Many economic reasoning tasks require precise multi-step calculations and correct interpretation of intermediate results. Small foundational errors—especially in exponent manipulation—can compound and degrade final outcomes. We investigated whether models that perform well on basic exponent tasks also perform better on downstream tasks that rely on them.

We first calibrated model performance on the Exponents element by normalizing it relative to the average accuracy across the benchmark. We then computed, for each downstream element, a performance gap defined as the difference between accuracy on tasks that involve exponentiation (those with Cobb–Douglas functions) and their linear counterparts that do not require exponent manipulation. Each (model, downstream element) pair is represented as a point in Figure 7, where the x-axis shows the performance gap and the y-axis displays the performance on the Exponents element.

The results reveal a clear trend: models stronger on Exponents exhibit smaller downstream gaps. This relationship is statistically significant (Pearson $r = 0.774$, $p = 1.23 \times 10^{-7}$), indicating that foundational skills explain a meaningful portion of performance variance. Notably, o3 achieves the best overall performance on the benchmark as well as on the Exponents element because it can rely on an internal code execution engine, avoiding manual errors.

## 6 Discussion and Conclusions

Our work introduces a novel benchmark specifically designed to evaluate LLMs' performance in non-strategic microeconomics, focusing on tasks that require a deep understanding of optimization, marginal analysis, and economic reasoning in individual decision-making contexts. This benchmark provides a comprehensive tool to assess the strengths and weaknesses of current models, revealing where they excel and where they struggle in applying foundational economic concepts. By identifying these areas, our benchmark can guide users in determining when LLMs can be trusted to perform well in economic analyses and when further development is needed.

In cases where models fall short, our benchmark serves as a practical resource for targeted improvements, e.g., via fine-tuning models, curating more specific datasets, or developing architectures better suited for microeconomic reasoning. These enhancements have the potential to impact a variety of economic applications, such as simulating consumer behavior, analyzing market dynamics, or conducting policy evaluations.

Looking ahead, we plan to expand our benchmark by incorporating additional elements from the microeconomics literature, deepening the evaluation of non-strategic decision-making. We encourage suggestions on new elements to include and make `auto-STEER` public for others to add more elements or expand on the elements we have currently. We also intend to explore further experimentation with additional LLMs, adaptation strategies, and prompt configurations, along with more detailed analyses of model performance.

## 7 Acknowledgements

This work was funded by an NSERC Discovery Grant, a DND/NSERC Discovery Grant Supplement, a CIFAR Canada AI Research Chair (Alberta Machine Intelligence Institute), awards from Facebook Research and Amazon Research, and DARPA award FA8750-19-2-0222, CFDA #12.910 (Air Force Research Laboratory). This paper draws on research supported by the Social Sciences and Humanities Research Council.

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

# A  Taxonomy of Non-Strategic Microeconomics

## A.1  DECISIONS ON CONSUMPTION IN NON-STRATEGIC ENVIRONMENTS

We begin by characterizing the space of elements that test an agent's ability to optimally allocate their limited resources to goods and services they desire. In economics and decision theory, the most primitive approach to describing the preferences of decision-makers is to use a function that maps a set of possible choices to the agent's optimal choice within that set. Under a set of intuitive assumptions, such as *transitivity* (i.e., if bundle $X$ is preferred to bundle $Y$, and $Y$ is preferred to bundle $Z$, then $X$ must be preferred to $Z$), it becomes possible to "rationalize" preferences by instead describing a utility function. This function assigns a real number to each bundle, and the agent selects the bundle with the highest utility.

In this paper, we focus on these "rationalizable" preferences, where agent choice can be implemented as utility maximization constrained by prices and income. The solution to these consumer choice problems provides us with, among other things, individual demand functions, which describe the choice of each good or service as a function of prices and income. The individual demand functions for each good are essential when aggregating to the market demand in Consumer Goods Market Aggregation, which in turn is used to find the price in a non-strategic equilibrium. In addition, we test variations on the framework such as the agents ability to make tradeoffs between the quantity of goods they would need to be able to purchase for an increase in the amount of work they provide for a given wage (i.e., the elasticity of labor supply), as well as cases of choice under uncertainty where the agent is choosing between possible lotteries under rationalizability assumptions required for von Neuman expected utility.

### A.1.1  Properties of Utility Functions

In this section, we test the ability of the agent to use utility functions as a means to compare preferences over different "bundles" of goods or services. A key feature of economic reasoning in this context is for agents to consider how substitution between different goods in a bundle might achieve the same utility (i.e., map out the "indifference curves"). Key tests include correctly distinguishing between substitutes and complements in consumption, and calculating the marginal rate of substitution at a point on an indifference curve. This logic is essential for both agents acting as a planner as we will see in Appendix A.4 and when fulfilling the role of choice under budget and income constraints, in Deriving Demand.

**Element A.1 (Marginal Utility).** *The ability to calculate marginal utility for different types of demand curves such as quasilinear, Cobb-Douglas, and Leontief.*

**Element A.2 (Diminishing Marginal Utility).** *The ability to recognize the role of diminishing marginal utility in consumption decisions and the role of achieving interior solutions.*

**Element A.3 (Marginal Rate of Substitution).** *The ability to calculate the marginal rate of substitution between two goods in a consumption decision.*

**Element A.4 (Tangency and the Marginal Rate of Substitution).** *The ability to calculate the marginal rate of substitution between two goods in a consumption decision at a given point in the budget constraint as tangent to the indifference curve.*

**Element A.5 (Substitutes and Complements).** *The ability to distinguish between substitutes and complements in consumption decisions.*

### A.1.2  Deriving Demand

The module in this section tests an agent's ability to solve a constrained utility maximization problem to derive a demand function—relying on the results of Properties of Utility Functions. We test the canonical classes of demand functions, check the duality of Marshallian demand and Hicksian demand, and ask the agent to derive these demand functions from first principles.

**Element A.6 (Derivation of Marshallian Demand).** *The ability to calculate the demand curve for a good given a utility function and a budget constraint.*

**Element A.7 (Derivation of Hicksian Demand from Expenditure Minimization).** *The ability to calculate the demand curve for a good given a utility function and a budget constraint.*

**Element A.8 (Duality of Hicksian Demand).** *The ability to recognize that Hicksian demand (expenditure minimization) is dual to maximization in Marshallian Demand.*

### A.1.3 Comparative Statics of Demand

This module considers how agents reason about changes in prices or income, and their effects on the quantity of each good they would purchase. We test the classic law of demand, different types of goods (e.g., normal, inferior, and Giffen), and derive Engel curves from first principles. The key tests are to ensure the agent rationally responds to changes in relative prices, and investigate their substitution between goods in a bundle. In practice, these tests involve comparative statics of the argmax from the utility maximization of the previous section on Deriving Demand—i.e., using an Envelope theorem and perturbing prices or income.

**Element A.9 (Law of Demand).** *The ability to calculate the change in demand with the change in price for normal goods.*

**Element A.10 (Price Elasticity of Demand).** *The ability to calculate the price elasticity of demand for a good given a utility function and a budget constraint.*

**Element A.11 (Consumption Changes).** *The ability to change the relative expenditures on goods given changes in relative prices with ordinary or Giffen goods.*

**Element A.12 (Engel Curves).** *The ability to calculate the Engel curve for a good given a utility function and a budget constraint.*

**Element A.13 (Income Elasticity of Demand).** *The ability to calculate the income elasticity of demand for a good given a utility function and a budget constraint.*

### A.1.4 Labor Supply

While the proceeding elements tested tradeoffs in choices of bundles with different goods, services (in Deriving Demand and over lotteries in Dynamic Consumption Decisions), often agents need to make a choice trading off between leisure and consumption. The elements in this module test an agent's ability to optimally make that tradeoff by balancing the consumption goods required to compensate for decreased leisure—which leads to the labor supply elasticity central to many branches of economics. Since goods must be purchased, agents will consider the relative wage from additional work compared to the price of goods. This leads us to be able to test an agent's ability to distinguish real from nominal prices.

**Element A.14 (Deriving Labor Supply).** *The ability to calculate the labor supply curve given specific preference parameterizations such as separable preferences or homothetic preferences.*

**Element A.15 (Labor Supply Elasticity).** *The ability to calculate the elasticity of labor supply.*

**Element A.16 (Marginal Rate of Substitution in Labor Supply).** *The ability to calculate the marginal rate of substitution between consumption and leisure in a labor supply decision.*

### A.1.5 Dynamic Consumption Decisions

Individuals often face decisions about how to trade off more consumption today at the cost of additional debt and less consumption in the future, and how best to plan for consumption with various contingencies with the future is uncertain. Among other applications, this provides a formal model of how to best choose a mixture of financial assets—i.e., portfolios. Consequently, this subsection tests intertemporal consumption choices, optimal portfolio choice—which involves selecting a mix of assets that maximizes expected utility given the risks and returns associated with each asset. Understanding portfolio choice helps explain how consumers manage risk and make investment decisions, which is vital for financial planning and economic stability.

**Element A.17 (Price of Risk with Mean-Variance Utility).** *The ability to calculate the price of risk for a mean-variance utility function.*

**Element A.18 (State-Contingent Consumption).** *The ability to calculate the optimal consumption given a utility function and a set of state-contingent consumption bundles.*

**Element A.19 (Arbitrage).** *The ability to recognize and execute arbitrage opportunities given two goods and prices you can resell.*

**Element A.20 (Optimal Portfolio Choice with Bid-Ask Spreads).** *The ability to calculate the optimal portfolio given bid-ask spreads.*

**Element A.21 (Exponential Discounting).** *The ability to exponentially discount future rewards or costs.*

**Element A.22 (Intertemporal Consumption Smoothing).** *The ability to calculate a smoothed consumption path and determine whether it is preferred to a non-smoothed path.*

## A.2 DECISIONS ON PRODUCTION IN NON-STRATEGIC ENVIRONMENTS

In the previous section, we derived how an agent facing a set of prices would choose the quantity demanded of each good or service to maximize their utility function. We also tested the amount of time that an agent might choose to wok (i.e., the quantity of labor supplied) given market wages—where the agent trades off the additional goods they might purchase against the lost leisure time they must forgo. Here, we look at the other side of the market and test an agent's ability to operate a production technology to maximize profits. Facing market prices for all production factors (e.g., wages and the capital) and the market price of the good or service they produce, the agent chooses the quantity of each factor of production and the total output. Parallel to DECISIONS ON CONSUMPTION IN NON-STRATEGIC ENVIRONMENTS, in Properties of Production Functions we first test general properties of production functions to ensure the agent can reason about substitution between factors, economies of scale in production, etc. Then in Deriving Factor Demand we solve the firms optimal profit maximization problem to determine the optimal choice of factors of production and output given a set of market prices. Finally, in Comparative Statics with Production we test the agents ability to reason about comparative statics on prices and their impact on factor demand and firm output.

### A.2.1 Properties of Production Functions

Production functions in these environments take continuous inputs of each factor, which lets us test an agent's ability to conduct marginal thinking when choosing the composition of inputs. For example, by knowing the hourly wage of hiring an additional worker, the additional output the worker might produce using the particular production process, and the price they can sell the firm's output, they can decide whether hiring the additional worker is profitable. In the absence of prices, this section tests basic decision making of the agent for understanding substitution between factor of production, marginal products for each input, and the understanding of the returns to scale of a production process.

**Element A.23 (Marginal Products).** *The ability to calculate separate marginal products for a production function with multiple inputs (e.g., labor and capital).*

**Element A.24 (Input Price Elasticity).** *The ability to calculate the responsiveness of output to a proportional change in a specific input's cost, holding all other inputs constant.*

**Element A.25 (Output Elasticity).** *The ability to calculate the output elasticity of an input in a production function.*

**Element A.26 (Elasticity of Substitution).** *The ability to calculate the marginal elasticity of substitution between inputs in a production function.*

**Element A.27 (Diminishing Marginal Products).** *The ability to calculate the diminishing marginal products for a production function with multiple inputs.*

**Element A.28 (Average and Marginal Costs).** *The ability to calculate average and marginal costs given a production function and input prices, and use them to determine scale.*

**Element A.29 (Returns to Scale).** *The ability to determine the proportional change in output resulting from a proportional change in all inputs in a production function.*

### A.2.2 Deriving Factor Demand

This module tests the agent's ability to act in the role of a profit maximizer in non-strategic situations where they take as given the price which they could sell goods they produce, and must pay for inputs to their production process at market rates (e.g., a competitive wage). Whereas in Deriving Demand, the agent was solving a utility maximization problem subject to a budget constraint, here they solve a profit maximization problem constrained by a production function. We test decisions on the quantity and composition of inputs, and the quantity of output for canonical production functions such as Cobb-Douglas and Leontief production functions given the agent's understanding of production

functions from Properties of Production Functions. The agent is asked to derive the factor demand functions from first principles from profit maximization and test their ability to reason with the dual cost-minimization formulation—analogous to the Hicksian vs. Marshallian demand of Deriving Demand.

**Element A.30 (Profit Maximization).** *The ability to calculate the optimal input bundle for a firm given a production function and input prices. Examples of given production functions: Cobb-Douglas, Leontief, Perfect Substitutes, CES production, CRS production, fixed costs.*

**Element A.31 (Expenditure Minimization).** *The ability to calculate the optimal input bundle for a firm given a production function and input prices.*

**Element A.32 (Duality of Profit Maximization and Expenditure Minimization).** *The ability to recognize that profit maximization is dual to expenditure minimization in production decisions and achieve consistent solutions.*

### A.2.3   Comparative Statics with Production

This module considers how agents reason about changes in the prices at which they can sell their goods, as well as changes in the costs of producing those goods. In particular, we can test how this affects their optimal choice of inputs to their production process (e.g., how many people to hire or robots to lease). We test comparative statics on the prices of inputs to the production function, changes to the underlying production technology, and substitution between goods for classic production functions such as Cobb-Douglas and Leontief. Analogous to the relationship between Deriving Demand and Comparative Statics of Demand, these tests involve comparative statics of the argmax from the profit maximization of Deriving Factor Demand—i.e., using an Envelope theorem and perturbing factor prices.

**Element A.33 (Price Elasticity of Supply).** *The ability to calculate the price elasticity of supply for a good given a production function and input prices.*

**Element A.34 (Shephard's Lemma).** *The ability to calculate factor demands given a cost function using the derivatives with respect to prices.*

**Element A.35 (Input Price Elasticity).** *The ability to calculate how the optimal input bundle changes with changes in input prices for a given production function.*

**Element A.36 (Total Factor Productivity).** *The ability to calculate total factor productivity given a production function and input prices*

### A.2.4   Dynamic Production Decisions

While Deriving Factor Demand tested the ability of agents to make static (i.e., within-period) decisions on the mix of input factors to maximize profits, many producer problems are inherently dynamic. For example, we can test if an agent can optimally choose the amount of capital to purchase given forecasts of future consumer demand and prices or choose how much to adjust the labor force in cases when labor is difficult to relocate due to frictions such as hiring and firing costs. Finally, agents are tested on their ability to make optimal entry and exit decisions based on their forecasted profits in an evolving market.

**Element A.37 (Dynamic Profit Maximization).** *The ability to calculate the optimal investment decision given a production function and input prices.*

**Element A.38 (Entry and Exit Decisions).** *The ability to calculate the optimal entry and exit decisions given a production function and fixed costs.*

### A.3   DECISIONS IN MULTI-AGENT NON-STRATEGIC ENVIRONMENTS

This setting tests the core logic of the relationship between supply-and-demand and prices, building on the tests of optimal behavior in appendix A.2 and appendix A.1. Economists refer to "general equilibrium" as the process where equilibrium prices and quantities emerge with a large number of non-strategic, price-taking market participants interact. Unlike the strategic models found in STEER, the assumption is that the market interactions that lead to this equilibrium occur through an unspecified process that clears markets (i.e., a "Walrasian auctioneer" or "invisible hand").

In particular, for non-strategic settings, all market participants take prices as given and choose the quantity demanded or supplied in each market. For example, consumers jointly decide on the quantity

demanded of goods and services given relative prices, and the quantity of labor supplied given a wage. Simultaneously, producers choose the quantity supplied of the good and the demand of each factor of production. With a large number of non-strategic market participants we can test the agents ability aggregate all of their supply and demand functions to calculate a market-level supply and demand. Finally, given the aggregated supply and demand functions for each market, we can test whether an agent can find the market clearing price where supply is equal to demand in equilibrium—given their internal model of all the market participants.

In this section, we organize by markets rather than by the role of a decision maker, as in the previous sections. For example, in the goods market we first ensure agents understand how individual demand functions from Deriving Demand aggregate to a market demand function for the good given a price, then that the agent understands how to aggregate the output from each producer at a given price from Comparative Statics with Production, and finally that the agent is able to calculate the price which would equate demand and supply and clear the market in a non-strategic setting. Factor markets are treated similarly.

Finally, given a system of equations that defines an equilibrium price we can perturb primitives (e.g., technological factors, distortions on decisions such as tax rates, or exogenous prices not determined in equilibrium) to see how the market clearing price would respond. That is an essential tool for agents to be able to reason about the impact of interventions and distortions in Appendix A.4.

### A.3.1 Consumer Goods Market Aggregation

The market clearing prices in general equilibrium arise from the separate market-level demand and supply curves, which sums the demand or supply across all market participants at a given price. Here we test the aggregation of demand functions derived from individual preferences, as in Deriving Demand and Comparative Statics of Demand, to a market demand function that summarizes the total quantity demanded across all agents at a given price. Central to the tests is to verify that the agent can aggregate the demands of market participants with heterogeneous preferences. On the other side of the market, we test if the agent can aggregate the "supply functions" resulting from the optimal choice of factors in Deriving Factor Demand and Comparative Statics with Production.

**Element A.39 (Aggregation of Consumer Demand).** *The ability to calculate the aggregate demand for a good given primitives of demand into expenditure shares.*

**Element A.40 (Aggregation of Offer Curve for the Good).** *The ability to calculate the aggregate supply of a good given primitives of supply into production functions.*

### A.3.2 Factor Market Aggregation

As with the case of the goods market in Consumer Goods Market Aggregation the market demand and supply for factors of production are essential to find the market clearing price. For example, we test whether the agent can aggregate the individual labor supply curve decisions from market participants who work at a particular wage, following Labor Supply, into a market labor supply curve. On the other side of the market, we test whether the agent can aggregate the labor demand in Deriving Factor Demand from producers into a market labor demand curve. The same tests are essential for all factors of production, including capital.

**Element A.41 (Aggregation of Labor Demand).** *The ability to calculate the aggregate demand for labor given primitives of demand into expenditure shares.*

**Element A.42 (Aggregation of Capital Demand).** *The ability to calculate the aggregate demand for capital given primitives of demand into expenditure shares.*

**Element A.43 (Aggregation of Labor Supply).** *The ability to calculate the aggregate supply of labor given primitives of supply into production functions.*

**Element A.44 (Aggregation of Fixed Factor Supply).** *The ability to calculate the aggregate supply of capital given primitives of supply into production functions.*

### A.3.3 Prices in Static Market Equilibrium

In this setting we test the agent's ability to reason about how prices emerge in non-strategic setting as a process of equating supply and demand, which in turn relies on their ability to aggregate those market demand functions from consumer and producer behavior.

More specifically, the core logic of general equilibrium is to find the equilibrium price by taking the aggregated demand and supply functions for each market and find the prices which would equate demand and supply. For example, the supply and demand functions for the good, as a function of the price, in Consumer Goods Market Aggregation; or the supply and demand functions for factors of production, as a function of factor prices in Factor Market Aggregation. This is done market by market, taking all other prices as given—which requires the agent reason through comparative statics of the solution to a system of equations while keeping everything else fixed.

**Element A.45 (Find Equilibrium Price).** *The ability to calculate the equilibrium prices given a production function and a demand function.*

**Element A.46 (Factor Shares in Equilibrium).** *The ability to calculate the factor shares in a competitive equilibrium given a production function and input prices.*

### A.3.4 Comparative Statics of Equilibrium Prices

Here, we test whether agents can reason about how prices and allocations (e.g., labor, capital, and goods) would respond to changes in the environment. The canonical tests are to see how changes in model primitives (e.g., productivity of the production process) or exogenous forces from outside the model (e.g., impact of weather), change the equilibrium price and allocations of labor, capital, etc. that would clear the market and equate demand and supply.

**Element A.47 (Comparative Statics with Total Factor Production Shocks).** *The ability to calculate how equilibrium prices change with changes in input prices for a Cobb-Douglas production function.*

**Element A.48 (Comparative Statics with Inelastic or Perfectly Elastic Supply).** *The ability to calculate how equilibrium prices change with changes in input prices for a production function with inelastic or perfectly elastic supply.*

### A.4 EVALUATING EQUILIBRIA AND EXTERNALITIES

In DECISIONS IN MULTI-AGENT NON-STRATEGIC ENVIRONMENTS, we tested an agents ability to reason about equilibrium prices and quantities arising from supply and demand decisions in a non-strategic setting. Although preferences were reflected in the underlying supply and demand functions themselves (i.e., utility maximization in the consumption decisions of DECISIONS ON CONSUMPTION IN NON-STRATEGIC ENVIRONMENTS and profit maximization in the production decisions of DECISIONS ON PRODUCTION IN NON-STRATEGIC ENVIRONMENTS), the equilibria in DECISIONS IN MULTI-AGENT NON-STRATEGIC ENVIRONMENTS do not necessarily reflect broader social preferences.

However, we can still ask whether the resulting "allocations" (i.e., the physical goods produced and how they are distributed to individuals, the amount of hours worked, and the physical capital installed) from the "invisible hand" in DECISIONS IN MULTI-AGENT NON-STRATEGIC ENVIRONMENTS compare to a alternative ways of allocating resources which may directly take social preferences into account. A central result of economics in non-strategic settings is that absent market imperfections and market power (i.e., when self-interested agents cannot directly manipulate prices because they are too small) the competitive equilibria of DECISIONS IN MULTI-AGENT NON-STRATEGIC ENVIRONMENTS typically yields the same allocations a benevolent planner might choose.

In this section, we consider how a social planner would evaluate the underlying welfare, efficiency, and inequality that comes about in non-strategic equilbria with prices derived from equating supply and demand. This leads to testing the ability of the agent to evaluate Pareto efficiency, consider the welfare theorems, evaluate Pigouvian externalities, and weigh the welfare impact of various market interventions which change the equilibria derived in DECISIONS IN MULTI-AGENT NON-STRATEGIC ENVIRONMENTS.

### A.4.1 Welfare and Decentralization

In this section, we test whether the agent can determine cases where the the competitive equilibrium they calculate would yield the same distribution of resources and consumer welfare as that of a benevolent social planner directly making the consumption and production decisions of all agents directly (also known as the "Welfare Theorems"). In cases where the supply-and-demand relationships lead to the same results as those of a planner, the competitive equilibrium and its prices are said to

"decentralize" the problem of a social planner. We then test that the agent recognizes cases where the welfare theorems fail, and can calculate the degree of welfare loss due to the distortions.

**Element A.49 (First Welfare Theorem).** *The ability to recognize that a competitive equilibrium is Pareto efficient.*

**Element A.50 (Second Welfare Theorem).** *The ability to recognize that any Pareto efficient allocation can be achieved as a competitive equilibrium with prices.*

**Element A.51 (Consumer Surplus).** *The ability to calculate consumer surplus given a demand curve and a price.*

**Element A.52 (Producer Surplus).** *The ability to calculate producer surplus given a supply curve and a price.*

**Element A.53 (EFficient Surplus).** *The ability to calculate the total surplus in a competitive equilibrium and recognize that it is maximized in the competitive equilibrium.*

**Element A.54 (Deadweight Loss of a Monopoly).** *The ability to calculate the deadweight loss of a monopoly given a demand curve and a supply curve.*

### A.4.2 Welfare Analysis of Market Equilibrium

In this section, we focus on the agent's ability to evaluate welfare implications of various forms of market equilibrium, particularly how different policies and distortions impact overall efficiency and resource allocation. The agent is tested on their understanding of how different interventions—such as taxes, subsidies, and price controls—affect welfare outcomes, and their ability to distinguish between distortionary and non-distortionary policies.

**Element A.55 (Identify Non-Distortionary Taxes).** *The ability to identify taxes which do not distort the allocation of resources.*

**Element A.56 (Irrelevance of Tax Incidence).** *The ability to recognize that the incidence of a tax does not depend on who is legally responsible for paying the tax.*

**Element A.57 (Labor Supply Distortions).** *The ability to determine the extent that labor taxes will distort labor supply and change aggregates and prices.*

**Element A.58 (Capital Market Distortions).** *The ability to identify that taxing a fixed factor is non-distortionary, but distorts with dynamic accumulation.*

## B  Mitigating Data Contamination with `auto-STEER`

Data contamination, where training data inadvertently includes information from test sets, poses significant challenges in machine learning, leading to overestimated model performance and compromised generalization capabilities. To address this, we implemented a structured dataset generation methodology incorporating human oversight, controlled data generation, and style transfer techniques. This appendix details our approach and its alignment with best practices in the literature.

The `auto-STEER` methodology provides a systematic approach to generating datasets that mitigates the risk of data contamination, ensuring the integrity of benchmarks and the validity of results. Below, we outline the key aspects of `auto-STEER` that address this issue:

### B.1  Challenging Models with Rephrasings:

Rephrasings are known to cause significant variance in model performance, as demonstrated in the GSM-Symbolic dataset Mirzadeh et al. [2024] and other studies [e.g., Zhu et al., 2024, Wang et al., 2023] highlighting how syntactic or stylistic changes can challenge generalization. In Appendix G, we also show that much of the observed variance in LLM performance arises from these rephrasings, underscoring their role in robust evaluations. `auto-STEER` leverages this phenomenon to craft diverse rephrased questions that test beyond rote learning.

### B.2  Dynamic Question Generation:

`auto-STEER` generates new questions through a structured process that balances diversity and consistency. Questions are systematically rephrased or style-transferred to ensure they are different

enough from the original templates to prevent memorization while retaining the same core meaning. This approach reduces the risk of overlap with pre-trained data while preserving the focus of the assessment.

The rapid advancement of large language models necessitates benchmarks that can evolve just as quickly. To address this, `auto-STEER` incorporates a user interface that allows users to regenerate entire datasets with minimal effort. By modifying domains, seeds, or even resampling numerical values, users can quickly produce an entirely new dataset with minimal effort. This adaptability ensures that benchmarks remain fresh and resistant to contamination as models advance.

## C    Technical Descriptions of Metrics

### C.1    Accuracy.

Accuracy is the most broadly used metric for evaluating LLMs. We define accuracy metrics as metrics that only look at the top token that the LLM outputs.

#### C.1.1    Exact-match accuracy

This is the fraction of questions answered correctly. In the FTQA format, we deem a LLM's response correct if its final answer, when rounded to the same number of significant figures as the model's output, matches the correct answer. This ensures that models are not rewarded for being vaguely correct at low precision, while also penalizing overprecision: if an LLM reports more significant figures than necessary and is incorrect, that discrepancy is treated as an error. This evaluation aligns correctness with both the accuracy and confidence implied by the LLM's numerical output.

#### C.1.2    Normalized accuracy

Elements can differ in their number of multiple choice options, leading to differences in the exact-match accuracy of random guessing. We can compensate for this by reporting the gap between the LLM's exact-match accuracy and random guessing [Budescu and Bar-Hillel, 1993]. We compute normalized accuracy for an element as follows: $\sum_{i=1}^{N} a_i(t_i) - \frac{1-a_i(t_i)}{|O_i|-1}$, where $t_i$ is the top token the LLM outputs for question $i$, $a_i$ is the indicator describing whether the top token is correct or not, $N$ the number of questions in the element, and $|O_i|$ the number of options in the question. In other words, normalized accuracy rewards an LLM with 1 point for every correct answer and penalizes an LLM by 1 over the number of options minus 1 for each incorrect answer.

### C.2    Calibration

It can also be useful to understand how confident an LLM is in its responses and the extent to which these confidence levels align with accuracy.

#### C.2.1    Expected calibration error

We follow Liang et al. [2022] and Raman et al. [2024] in measuring the confidence of an LLM's response and computing the expected calibration error [ECE; Naeini et al., 2015, Guo et al., 2017]. ECE measures how closely the probability an LLM assigns to its top answer matches the actual probability of the correct answer, which in our case is 1. ECE first splits the data into $M$ equally spaced bins, where each bin contains the probabilities the model assigned to their top token in that range: e.g., let $p^{\max}$ be the set of most probable tokens for each question then if $M = 2$, then the first bin $B_1 = \{p \mid p \in p^{\max} \text{ and } p \in [0, 0.5]\}$. It is then defined as $\sum_{i \in [M]} |B_i|/N \cdot |\texttt{acc}(B_i) - \texttt{conf}(B_i)|$, where $\texttt{conf}(B_i)$ is the average probability the LLM assigned to its top token in bin $B_i$, and $\texttt{acc}(B_i)$ denotes the exact-match accuracy in bin $B_i$. We allow users to choose the number of bins, however, we set $M = 10$ uniformly spaced over the interval $[0, 1]$ as is standard.

### C.2.2 Brier score

The Brier Score of an element is defined as

$$\sum_{i=1}^{N} \frac{1}{|O_i|} \sum_{o \in O_i} (p_i(o) - a_i(o))^2,$$

where $p_i(o)$ is the probability the LLM assigns to option $o$ in question $i$. Thus, if an LLM is overly confident in an incorrect answer (e.g., assigns a probability of 0.9 to a wrong option), the Brier Score will penalize it more heavily.

### C.2.3 Expected probability assignment

EPA measures how much probability mass an LLM assigns to the correct answer option out of all possible options. It is defined as: $1/N \sum_{i \in [N]} p_i^*$, where $p_i^*$ is the predicted probability that the LLM assigns to the correct option for question $i$.

## D  Technical Descriptions of Functional Families

In this section, we describe the functional forms that we use in testing economic concepts. Each can be applied to *consumer* problems (as utility functions) or *producer* problems (as production functions). We highlight the canonical mathematical form and note any technical differences in interpretation when modeling consumers versus producers.

### D.1  Cobb-Douglas

The *Cobb-Douglas* functional form is one of the most frequently used due to its tractable properties and partial elasticities interpretation. Suppose there are $n$ goods (or inputs). For a producer with input vector $\mathbf{x} = (x_1, x_2, \ldots, x_n)$, a typical Cobb-Douglas production function can be written as:

$$f(\mathbf{x}) = A \, x_1^{\alpha_1} \, x_2^{\alpha_2} \, \cdots \, x_n^{\alpha_n},$$

where $A > 0$ is a scale parameter and each $\alpha_i \geq 0$. For a consumer's utility function, the same functional family looks like:

$$u(\mathbf{q}) = q_1^{\beta_1} \, q_2^{\beta_2} \, \cdots \, q_n^{\beta_n},$$

where $\mathbf{q} = (q_1, q_2, \ldots, q_n)$ are quantities of goods consumed, and $\beta_i \geq 0$. Economically, $\alpha_i$ (or $\beta_i$) often reflect the relative importance (or expenditure share) of each input (or good).

### D.2  Leontief

A *Leontief* functional form encodes strict complementarity. A producer's Leontief production function is described as:

$$f(\mathbf{x}) = \min\left\{ \frac{x_1}{a_1}, \ \frac{x_2}{a_2}, \ \ldots, \ \frac{x_n}{a_n} \right\},$$

where each $a_i > 0$ captures a fixed proportion in which inputs must be combined. For a consumer, their Leontief utility function is of the form:

$$u(\mathbf{q}) = \min\left\{ \frac{q_1}{\gamma_1}, \ \frac{q_2}{\gamma_2}, \ \ldots, \ \frac{q_n}{\gamma_n} \right\}.$$

This implies goods are perfect complements: the consumer gains utility only when goods are consumed in the specific ratio $\gamma_1 : \gamma_2 : \ldots : \gamma_n$. In production, perfect complementarity imposes that a shortage of any one input strictly limits total output.

### D.3  Linear

The *linear* family is the simplest and assumes perfect substitutability. For a producer, the linear production function with inputs $\mathbf{x}$ takes the form:

$$f(\mathbf{x}) = b_1 x_1 + b_2 x_2 + \cdots + b_n x_n,$$

where $b_i \geq 0$. This means each input contributes additively (and independently) to output. A consumer's linear utility function with goods $\mathbf{q}$ is:

$$u(\mathbf{q}) \;=\; \theta_1 q_1 + \theta_2 q_2 + \cdots + \theta_n q_n,$$

where $\theta_i > 0$ captures the marginal utility for good $i$. In both contexts, linear forms imply a constant rate of technical (or preferential) substitution, reflecting strong substitutability among inputs (or goods).

## D.4 Non-Linear

Beyond the classic forms, we use low-degree polynomial specifications for both production and utility functions. Such functions can capture nonlinearities without resorting to strictly Cobb-Douglas or other functional families. For a producer with inputs $\mathbf{x}$, a second-degree polynomial looks like:

$$f(\mathbf{x}) \;=\; \alpha_0 \;+\; \sum_{i=1}^{n} \alpha_i\, x_i \;+\; \sum_{i=1}^{n}\sum_{j=1}^{n} \alpha_{ij}\, x_i\, x_j,$$

where $\alpha_0, \alpha_i, \alpha_{ij}$ are specified parameters. Similarly, for a consumer's utility function, we write:

$$u(\mathbf{q}) \;=\; \beta_0 \;+\; \sum_{i=1}^{n} \beta_i\, q_i \;+\; \sum_{i=1}^{n}\sum_{j=1}^{n} \beta_{ij}\, q_i\, q_j.$$

These forms allow more flexible curvature than linear or Cobb-Douglas, but remain computationally tractable for low degrees. The main difference in interpretation is that producers typically focus on *return-to-scale* questions or *marginal products*, while consumers focus on *marginal utilities* and *marginal rates of substitution*.

# E   Examples

## E.1   GPT-4o Aggregation of Consumer Demand (Shown)

**User Message:**

Q: As the owner of a premium bedding brand, I cater to a diverse group of customers. I've discovered that there are 1204 potential buyers for my luxury bedding products. Out of these, 820 shoppers in urban markets each follow the demand function $18.1 - 1.51P$, while 384 shoppers in suburban markets each adhere to the demand function $75.44 - 8.68P$. How can I calculate the total demand for my bedding at a price point of $8.4$?
A.  2779.36
B.  1832.01
C.  5411.87 *(Correct Answer)*
D.  5050.03
Let's think step by step. Explain your reasoning.

**GPT-4o Message:**

To calculate the total demand for your luxury bedding products at a price point of $8.4$, we need to evaluate the demand functions for both urban and suburban markets separately and then sum the results.

**Step 1: Calculate the demand in the urban market**

The demand function for the urban market is given by:

$$Q_u = 18.1 - 1.51P$$

Substitute $P = 8.4$ into the urban demand function:

$$Q_u = 18.1 - 1.51 \times 8.4$$

$$Q_u = 18.1 - 12.684$$

$$Q_u = 5.416$$

Since there are 820 shoppers in the urban market, the total demand in the urban market is:

$$\text{Total Urban Demand} = 820 \times 5.416$$

$$\text{Total Urban Demand} = 4441.12$$

**Step 2: Calculate the demand in the suburban market**

The demand function for the suburban market is given by:

$$Q_s = 75.44 - 8.68P$$

Substitute $P = 8.4$ into the suburban demand function:

$$Q_s = 75.44 - 8.68 \times 8.4$$

$$Q_s = 75.44 - 72.912$$

$$Q_s = 2.528$$

Since there are 384 shoppers in the suburban market, the total demand in the suburban market is:

$$\text{Total Suburban Demand} = 384 \times 2.528$$

$$\text{Total Suburban Demand} = 970.88$$

**Step 3: Calculate the total demand**

Now, sum the total demand from both markets:

$$\text{Total Demand} = \text{Total Urban Demand} + \text{Total Suburban Demand}$$

$$\text{Total Demand} = 4441.12 + 970.88$$

$$\text{Total Demand} = 5412$$

Therefore, the total demand for your bedding at a price point of $8.4$ is approximately $5411.87$.
The correct answer is:
C. 5411.87

## E.2   GPT-4o Aggregation of Consumer Demand (Hidden)

**User Message:**

Q: As the owner of a premium bedding brand, I cater to a diverse group of customers. I've discovered that there are 1204 potential buyers for my luxury bedding products. Out of these, 820 shoppers in urban markets each follow the demand function $18.1 - 1.51P$, while 384 shoppers in suburban markets each adhere to the demand function $75.44 - 8.68P$. How can I calculate the total demand for my bedding at a price point of $8.4$?
Let's think step by step. Explain your reasoning.

**GPT-4o Message:**

To calculate the total demand for your luxury bedding at a price point of 8.4, we need to evaluate the demand functions for both urban and suburban markets separately and then sum the results. Here's a step-by-step breakdown:

1. **Identify the demand functions:**
- Urban market demand function: $Q_u = 18.1 - 1.51P$
- Suburban market demand function: $Q_s = 75.44 - 8.68P$

2. **Substitute the price point (P = 8.4) into each demand function:**
- For the urban market:
$$Q_u = 18.1 - 1.51 \times 8.4$$

Calculate the value:
$$Q_u = 18.1 - 1.51 \times 8.4 = 18.1 - 12.684 = 5.416$$

- For the suburban market:
$$Q_s = 75.44 - 8.68 \times 8.4$$

Calculate the value:
$$Q_s = 75.44 - 8.68 \times 8.4 = 75.44 - 72.912 = 2.528$$

3. **Sum the demands from both markets to get the total demand:**
$$Q_{total} = Q_u + Q_s = 5.416 + 2.528 = 7.944$$

Therefore, at a price point of 8.4, the total demand for your luxury bedding products from both urban and suburban markets is approximately 7.944 units.

---

**User Message:**

A. 2779.36
B. 1832.01
C. 5411.87 *(Correct Answer)*
D. 5050.03
Answer by writing the option letter corresponding to the correct option. WRITE ONLY A SINGLE LETTER.
A:

---

**GPT-4o Message:**

B

---

### E.3 GPT-4o Profit Maximization

**User Message:**

Q: John manages a company that produces premium leather wallets. The production output depends on the capital (K) he invests, represented by the function $3.97K^{0.45}$. At the moment, the wallets are sold at a price of 3.64, and John has an initial capital of $K_1 = 3.3$. He needs to decide how much additional capital to invest for the next day's market. John is aware of the potential price distribution for his wallets for the next day, represented by price 2.7 with probability 0.24, price 2.14 with probability 0.42, price 3.52 with probability 0.34. There is also a cost associated with increasing his capital, given by $(K_1 - K_2)^2$, and he applies a discount factor of 0.37 to future revenues. To maximize his profits, how much additional capital should John invest?
A. 0.44
B. 0.14
C. 1.1
D. 0.09
Let's think step by step. Explain your reasoning.

**GPT-4o Message:**

To determine how much additional capital John should invest to maximize his profits, we need to consider the expected revenue from the different potential prices, the cost of increasing capital, and the discount factor applied to future revenues. Let's break this down step by step.

**Step 1: Calculate the Expected Revenue**

First, we need to calculate the expected price of the wallets for the next day using the given probabilities.

$$\text{Expected Price} = (2.7 \times 0.24) + (2.14 \times 0.42) + (3.52 \times 0.34)$$

$$\text{Expected Price} = 0.648 + 0.8988 + 1.1968 = 2.7436$$

**Step 2: Determine the Production Function**

The production function is given by $3.97K^{0.45}$. We need to calculate the production output for different levels of capital $K$.

**Step 3: Calculate the Revenue for Different Capital Levels**

Let's denote the initial capital as $K_1 = 3.3$. We need to consider the additional capital $\Delta K$ that John might invest. The new capital $K_2$ will be $K_1 + \Delta K$.

The revenue function for a given capital $K$ is:

$$\text{Revenue} = \text{Expected Price} \times \text{Production Output}$$

$$\text{Revenue} = 2.7436 \times 3.97K^{0.45}$$

**Step 4: Calculate the Cost of Increasing Capital**

The cost of increasing capital is given by $(K_1 - K_2)^2$. Since $K_2 = K_1 + \Delta K$, the cost becomes:

$$\text{Cost} = (\Delta K)^2$$

**Step 5: Calculate the Discounted Future Revenue**

John applies a discount factor of 0.37 to future revenues. Therefore, the discounted future revenue is:

$$\text{Discounted Revenue} = 0.37 \times \text{Revenue}$$

**Step 6: Calculate the Net Profit**

The net profit is the discounted revenue minus the cost of increasing capital:

$$\text{Net Profit} = 0.37 \times (2.7436 \times 3.97K_2^{0.45}) - (\Delta K)^2$$

**GPT-4o Message (Continued):**

**Step 7: Evaluate the Net Profit for Different Values of $\Delta K$**

We need to evaluate the net profit for the given options of additional capital investment:

Option A: $\Delta K = 0.44$

$$K_2 = 3.3 + 0.44 = 3.74$$
$$\text{Production Output} = 3.97 \times 3.74^{0.45}$$
$$\text{Revenue} = 2.7436 \times 3.97 \times 3.74^{0.45}$$
$$\text{Discounted Revenue} = 0.37 \times \text{Revenue}$$
$$\text{Cost} = (0.44)^2 = 0.1936$$
$$\text{Net Profit} = 0.37 \times \text{Revenue} - 0.1936$$

Option B: $\Delta K = 0.14$

$$K_2 = 3.3 + 0.14 = 3.44$$
$$\text{Production Output} = 3.97 \times 3.44^{0.45}$$
$$\text{Revenue} = 2.7436 \times 3.97 \times 3.44^{0.45}$$
$$\text{Discounted Revenue} = 0.37 \times \text{Revenue}$$
$$\text{Cost} = (0.14)^2 = 0.0196$$
$$\text{Net Profit} = 0.37 \times \text{Revenue} - 0.0196$$

Option C: $\Delta K = 1.1$

$$K_2 = 3.3 + 1.1 = 4.4$$
$$\text{Production Output} = 3.97 \times 4.4^{0.45}$$
$$\text{Revenue} = 2.7436 \times 3.97 \times 4.4^{0.45}$$
$$\text{Discounted Revenue} = 0.37 \times \text{Revenue}$$
$$\text{Cost} = (1.1)^2 = 1.21$$
$$\text{Net Profit} = 0.37 \times \text{Revenue} - 1.21$$

Option D: $\Delta K = 0.09$

$$K_2 = 3.3 + 0.09 = 3.39$$
$$\text{Production Output} = 3.97 \times 3.39^{0.45}$$
$$\text{Revenue} = 2.7436 \times 3.97 \times 3.39^{0.45}$$
$$\text{Discounted Revenue} = 0.37 \times \text{Revenue}$$
$$\text{Cost} = (0.09)^2 = 0.0081$$
$$\text{Net Profit} = 0.37 \times \text{Revenue} - 0.0081$$

**Step 8: Compare the Net Profits**

We need to compare the net profits for each option to determine which one maximizes John's profit. After calculating the net profits for each option, we find that:
- Option A: $\Delta K = 0.44$ yields the highest net profit.
Therefore, the best option for John is to invest an additional capital of 0.44.
**Answer: A. 0.44**

---

**User Message:**

Answer by writing the option letter corresponding to the correct option. WRITE ONLY A SINGLE LETTER.
A:

---

**GPT-4o Message:**

A

## F   Figures

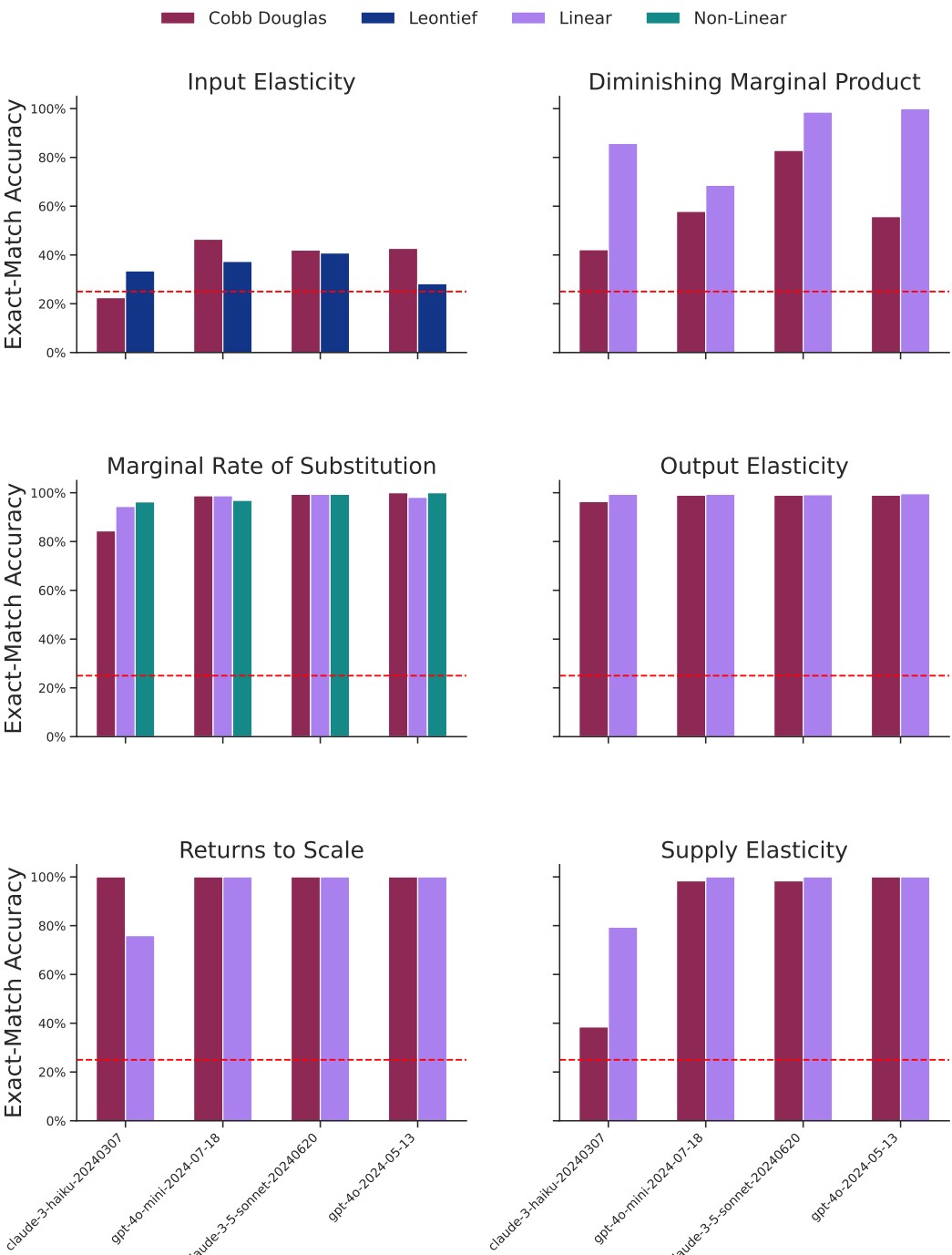

Figure 5: Exact-match accuracy of all closed-source models across six elements and four functional families (Cobb-Douglas, Leontief, Linear, and Non-Linear). The results demonstrate varying type robustness, with Cobb-Douglas being a generally more challenging functional family but not consistently harder for all elements. For instance, accuracy remains high for elements such as Output Elasticity and Marginal Rate of Substitution, even on the Cobb-Douglas functions, while elements like Input Price Elasticity and Returns to Scale show more variability across functional types. The red dashed line indicates the random guessing baseline for comparison.

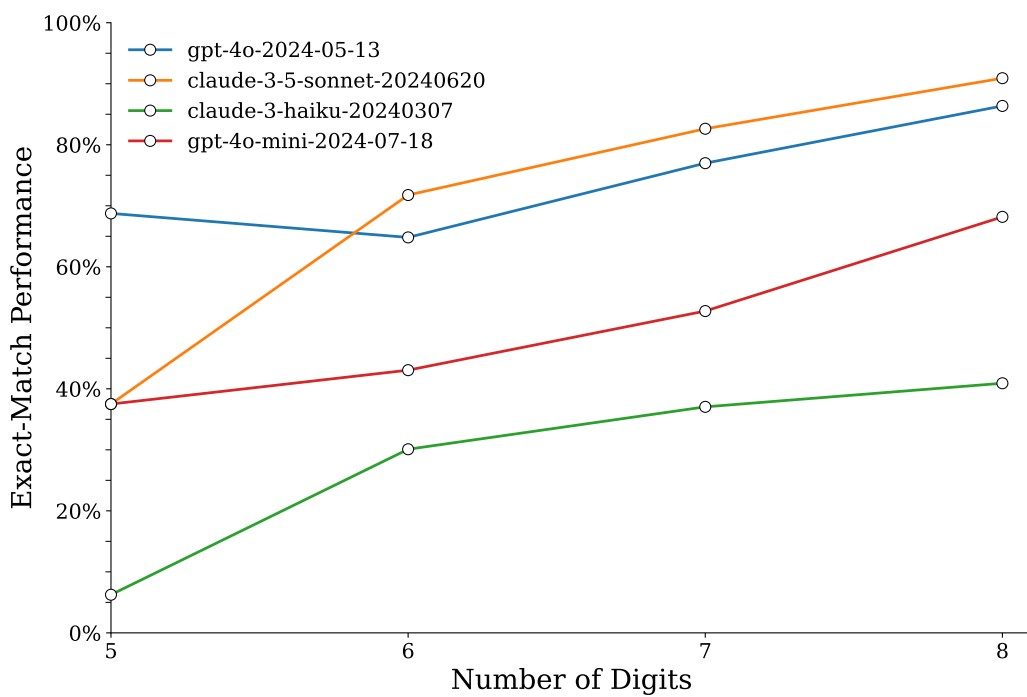

Figure 6: This figure depicts exact-match MCQA performance on the Aggregation of Consumer Demand element for the closed-source non-reasoning models against the number of digits of the correct answer.

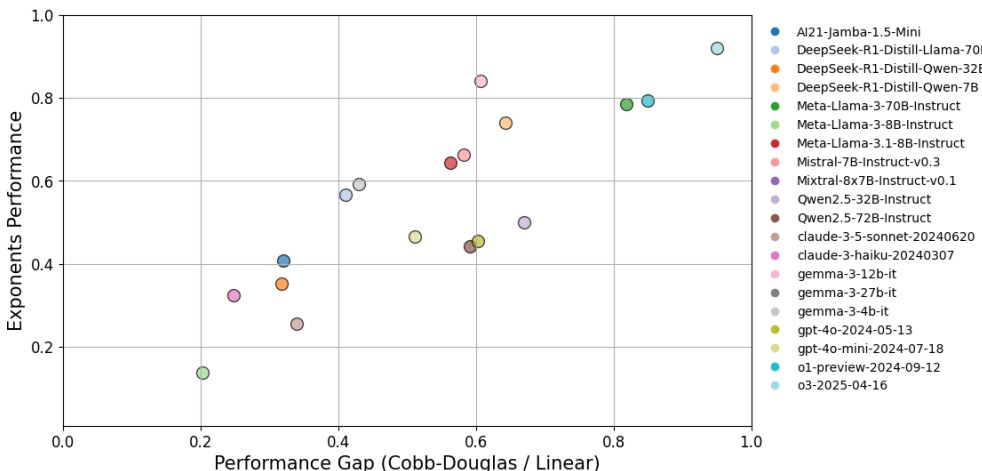

Figure 7: Scatter plot of calibrated performance on the Exponents element versus downstream performance gap across LLMs. The x-axis shows the gap calculated as the quotient between a LLM's accuracy on real-valued exponent-based (Cobb–Douglas) tasks and its accuracy on the linear version of those tasks for various downstream elements. The y-axis represents the LLM's performance on Exponents normalized by dividing by its average accuracy on the benchmark. Each point corresponds to a specific (LLM, downstream element) pair, with colors distinguishing different LLMs.

# G   Analysis of Rephrasing Variance

To understand the role of question rephrasings in our dataset, we conducted an analysis of variance (ANOVA) on all other controllable features. These features include type, domain, and perspective. The goal of this analysis was to quantify the variance in LLM performance attributable to these features and, by exclusion, infer the contribution of rephrasings to the remaining unexplained variance.

The results for the top-performing models, summarized in Table 2 through Table 6, indicate that the explained variance attributable to the controlled features is consistently low across all evaluated models. This leaves approximately $56\%$ (for claude-3-5-sonnet) and up to $91\%$ (for o1-preview) of the variance unexplained by the features included in the analysis. Given that question rephrasings are a systematic element of our dataset design and were not included as a feature in this analysis, we infer that the majority of this residual variance is due to differences in how models respond to semantically equivalent but syntactically varied prompts.

| Factor | Sum of Squares | Degrees of Freedom | F-Statistic | p-value |
|---|---|---|---|---|
| domain | 7.5572 | 11.0000 | 3.0199 | 0.0823 |
| perspective | 3.4351 | 5.0000 | 3.0199 | 0.0823 |
| CAR | 21.1235 | 1.0000 | 92.8524 | 0.0000 |
| element:type | 178.6251 | 260.0000 | 3.0199 | 0.0823 |
| Residual | 2218.5356 | 9752.0000 | | |
| | *R-squared* | | | 0.0941 |
| | *Adjusted R-squared* | | | 0.0907 |

Table 2: ANOVA Results for o1-preview-2024-09-12

| Factor | Sum of Squares | Degrees of Freedom | F-Statistic | p-value |
|---|---|---|---|---|
| domain | 0.2619 | 11.0000 | 0.0928 | 0.7607 |
| perspective | 0.1190 | 5.0000 | 0.0928 | 0.7607 |
| 0-CoT | 229.7295 | 1.0000 | 895.0635 | 0.0000 |
| CAR | 141.3015 | 1.0000 | 550.5338 | 0.0000 |
| element:type | 47.4945 | 1995.0000 | 0.0928 | 0.7607 |
| Residual | 16015.5015 | 62399.0000 | | |
| | *R-squared* | | | 0.3368 |
| | *Adjusted R-squared* | | | 0.3358 |

Table 3: ANOVA Results for gpt-4o-2024-05-13

| Factor | Sum of Squares | Degrees of Freedom | F-Statistic | p-value |
|---|---|---|---|---|
| domain | 0.4573 | 11.0000 | 0.1293 | 0.7191 |
| perspective | 0.2079 | 5.0000 | 0.1293 | 0.7191 |
| 0-Cot | 34.0324 | 1.0000 | 105.8911 | 0.0000 |
| CAR | 159.1899 | 1.0000 | 495.3161 | 0.0000 |
| element:type | 82.9338 | 1995.0000 | 0.1293 | 0.7191 |
| Residual | 18642.5829 | 58006.0000 | | |
| | *R-squared* | | | 0.2964 |
| | *Adjusted R-squared* | | | 0.2953 |

Table 4: ANOVA Results for gpt-4o-mini-2024-07-18

| Factor | Sum of Squares | Degrees of Freedom | F-Statistic | p-value |
|---|---|---|---|---|
| domain | 1.2389 | 11.0000 | 0.5004 | 0.6063 |
| perspective | 0.5032 | 5.0000 | 0.4472 | 0.5037 |
| 0-CoT | 30.9486 | 1.0000 | 137.5118 | 0.0000 |
| CAR | 156.8135 | 1.0000 | 696.7577 | 0.0000 |
| element:type | 224.0990 | 1995.0000 | 0.4991 | 0.6071 |
| Residual | 18025.1941 | 80090.0000 | | |
| | | *R-squared* | | 0.4436 |
| | | *Adjusted R-squared* | | 0.4430 |

Table 5: ANOVA Results for claude-3-5-sonnet-20240620

| Factor | Sum of Squares | Degrees of Freedom | F-Statistic | p-value |
|---|---|---|---|---|
| domain | 1.4703 | 11.0000 | 0.3814 | 0.5369 |
| perspective | 0.6683 | 5.0000 | 0.3814 | 0.5369 |
| 0-Cot | 0.2629 | 1.0000 | 0.7502 | 0.3864 |
| CAR | 0.4909 | 1.0000 | 1.4007 | 0.2366 |
| element:type | 266.6568 | 1995.0000 | 0.3814 | 0.5369 |
| Residual | 38842.9909 | 110826.0000 | | |
| | | *R-squared* | | 0.2336 |
| | | *Adjusted R-squared* | | 0.2330 |

Table 6: ANOVA Results for claude-3-haiku-20240307

# H Models

| Model Name | Model Card | Chat/ Instruction Tuned |
|---|---|---|
| **Closed-Source** | | |
| *OpenAI* | | |
| o3 | | ✓ |
| o1-preview | | ✓ |
| gpt-4o | | ✓ |
| gpt-4o mini | | ✓ |
| *Anthropic* | | |
| claude-3-5-sonnet | | ✓ |
| claude-3-haiku | | ✓ |
| **Open-Source** | | |
| *Google* | | |
| gemma-3-4b-it | google/gemma-3-4b-it | ✓ |
| gemma-3-12b-it | google/gemma-3-12b-it | ✓ |
| gemma-3-27b-it | google/gemma-3-27b-it | ✓ |
| *Qwen* | | |
| Qwen2.5-72B-Instruct | Qwen/Qwen2.5-72B-Instruct | ✓ |
| Qwen2.5-7B-Instruct | Qwen/Qwen2.5-7B-Instruct | ✓ |
| Qwen2.5-3B-Instruct | Qwen/Qwen2.5-3B-Instruct | ✓ |
| Qwen2.5-0.5B-Instruct | Qwen/Qwen2.5-0.5B-Instruct | ✓ |
| Qwen2.5-math-7B-Instruct | Qwen/Qwen2.5-math-7B-Instruct | ✓ |
| Qwen2.5-math-1.5B-Instruct | Qwen/Qwen2.5-math-1.5B-Instruct | ✓ |

| Model Name | Model Card | Chat/Instruction Tuned |
|---|---|---|
| *Meta Llama* | | |
| Llama-3-8B-Instruct | meta-llama/Meta-Llama-3-8B-Instruct | ✓ |
| Llama-3-70B-Instruct | meta-llama/Meta-Llama-3-70B-Instruct | ✓ |
| Llama-3.1-8B | meta-llama/Meta-Llama-3.1-8B | ✗ |
| Llama-3.1-70B-Instruct | meta-llama/Meta-Llama-3.1-70B-Instruct | ✓ |
| *Mistral* | | |
| Mixtral-8x7B-Instruct-v0.1 | mistralai/Mixtral-8x7B-Instruct-v0.1 | ✓ |
| Mistral-7B-Instruct-v0.3 | mistralai/Mistral-7B-Instruct-v0.3 | ✓ |
| *AI21* | | |
| AI21-Jamba-1.5-Mini | ai21labs/AI21-Jamba-1.5-Mini | ✗ |

Table 7: Overview of the open- and closed-source LLMs we evaluated. The table includes their names, their model card links, and whether they have been chat or instruction tuned. Models are grouped by family and sorted by parameter size, with non-chat-tuned models listed first within each group.

# I   Extra Results

## I.1   Performance on Elements Generated by Claude 3.5 Sonnet

To assess whether the performance on our dataset was influenced by the choice of the generation LLM, we re-generated three elements from scratch using claude-3-5-sonnet. We selected Find Equilibrium Price because it exhibited the largest performance gap between gpt-4o and claude-3-5-sonnet, Diminishing Marginal Products was chosen as a random element with slight performance variation across the models, and Price Elasticity of Demand served as a control where no significant differences were expected.

Figure 8 shows the exact-match performance of both models on these three elements. We found no significant differences in performance between any of the models.

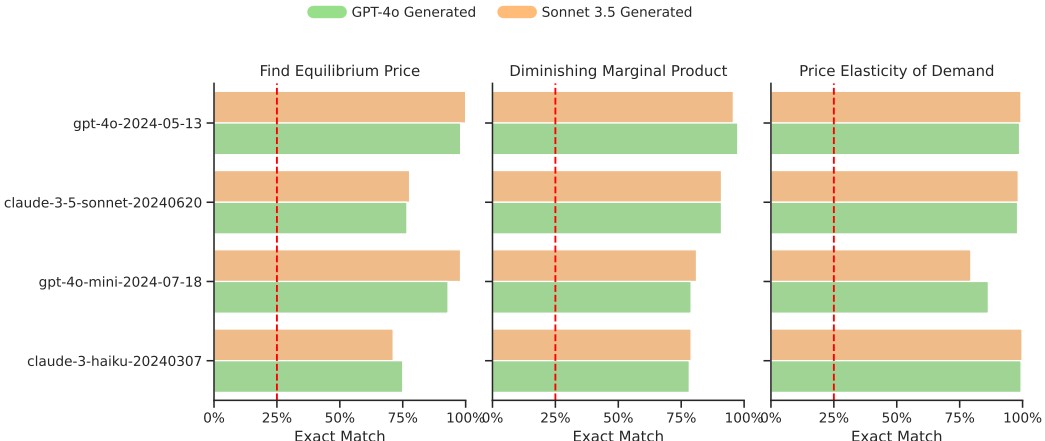

Figure 8: Exact-match performance comparison between closed-source models on three that were generated by gpt-4o and claude-3-5-sonnet. The elements were selected based on the observed performance differences across models, with Price Elasticity of Demand serving as a control. The figure shows no significant differences in performance between the models on these elements. Note that the red dotted line signifies random guessing performance.

## I.2 Intertemporal Consumption Smoothing

When optimizing intertemporal consumption, the consumer maximizes the discounted utility

$$\sum_{t=0}^{T} \beta^t u(c_t)$$

subject to the intertemporal budget constraint. The first-order condition for an optimum leads to the Euler equation:

$$u'(c_t) = \beta(1 + r)\, u'(c_{t+1}).$$

For our purposes, we tested models using a constant relative risk aversion (CRRA) utility function. We used the following form:

$$u(c) = \frac{c^{1-\gamma}}{1 - \gamma} \quad (\gamma \neq 1),$$

where $\gamma$ is the coefficient of relative risk aversion. This function exhibits diminishing marginal utility, meaning that each additional unit of consumption adds less to utility when overall consumption is high compared to when it is low. Due to diminishing returns, an agent is motivated to smooth consumption over time, even with a discount factor of 1; spending too much in one period reduces the marginal utility in that period, while having too little in another period results in a steep loss of satisfaction.

For CRRA utility, the Euler equation is given by:

$$u'(c_t) = \beta(1 + r)\, u'(c_{t+1}),$$

where

$$u'(c) = c^{-\gamma}.$$

Rearranging yields:

$$c_{t+1} = [\beta(1 + r)]^{\frac{1}{\gamma}}\, c_t.$$

However, we found that LLMs would often use linear utility functions in their analysis. For a linear utility function, the Euler equation—which equates the marginal benefit of consuming today with that of consuming tomorrow—simplifies significantly. If $u'(c)$ is constant (say, equal to 1), then aside from the effects of discounting and interest, there is no curvature-driven motive to adjust consumption levels across periods. The optimal allocation would then depend solely on the intertemporal budget constraint and the returns on savings.

## I.3 Deadweight Loss

To conduct our error analysis, we ran all closed-source models on the free-text QA adaptation of the deadweight loss task. We began by inspecting a range of model outputs to identify distinct classes of errors that were common across responses. Once these error categories were established, we computed the answers corresponding to these errors and rescored the models based on whether their outputs were within 98 % of either the correct answer or any of the answers derived from specific error assumptions. We also ensured that when there was any overlap in incorrect responses that we chose the closest one to the model's response. This approach allowed us to capture not only the frequency of correct outputs but also the systematic nature of the models' reasoning flaws. Below, we provide a detailed breakdown of the primary error types:

- Incorrect Base for Deadweight Loss Type 1: This error incorrectly substitutes $P_e - P_m$ (the difference between the competitive equilibrium price and the monopolist's price) in place of the correct term $P_m - MC(Q_m)$ (the difference between the monopolist's price and the marginal cost at the monopolist's quantity).
- Incorrect Base for Deadweight Loss Type 2: This error calculates the deadweight loss using the difference between the monopoly price and the competitive equilibrium price as the base of the triangle.
- Incorrect Base and Height Type 1: This error replaces the base of the DWL triangle ($Q_e - Q_m$) with a miscalculated value for the equilibrium quantity and replaces the base with the Type 1 variant.

- Incorrect Base and Height Type 2: This error assumes that $P_e = MC(Q_m)$, leading to an incorrect height calculation where the DWL triangle's height becomes $P_m - P_e$.
- Treating Marginal Cost as Constant: Instead of recognizing marginal cost as a function derived from the supply curve ($MC(Q) = a_{\text{supply}} \times Q + b_{\text{supply}}$), several models treated marginal cost as a constant, often equal to the slope of the supply curve. This assumption led to errors in determining the monopolist's quantity and price, further propagating inaccuracies in the deadweight loss calculation.
- Combined Errors: A subset of models combined the two errors above, simultaneously using an incorrect formula for deadweight loss and assuming a constant marginal cost. This compounded error significantly reduced the likelihood of producing a correct answer and highlighted the systematic nature of the misunderstanding.

# J  Section 3 Images

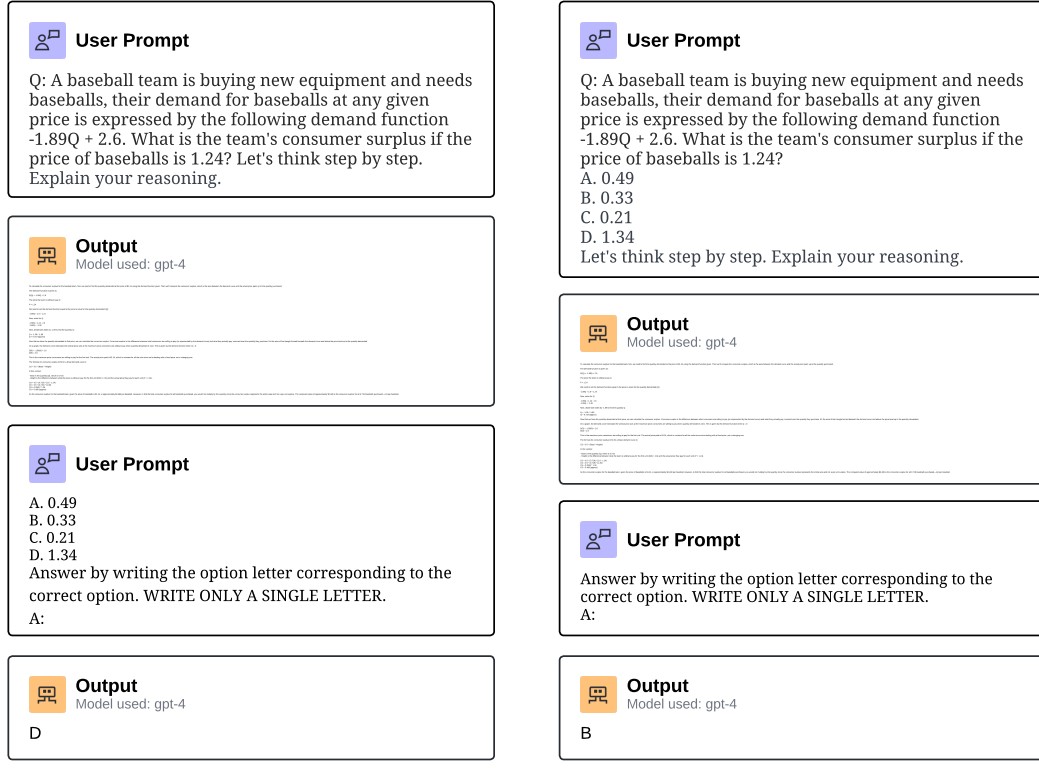

Figure 9: (Left) The hidden approach to 0-CoT: the model is given only the question and asked to explain its reasoning before being provided with options. (Right) The shown approach to 0-CoT: the model is presented with both the question and options before explaining its reasoning.

# K   Web Application

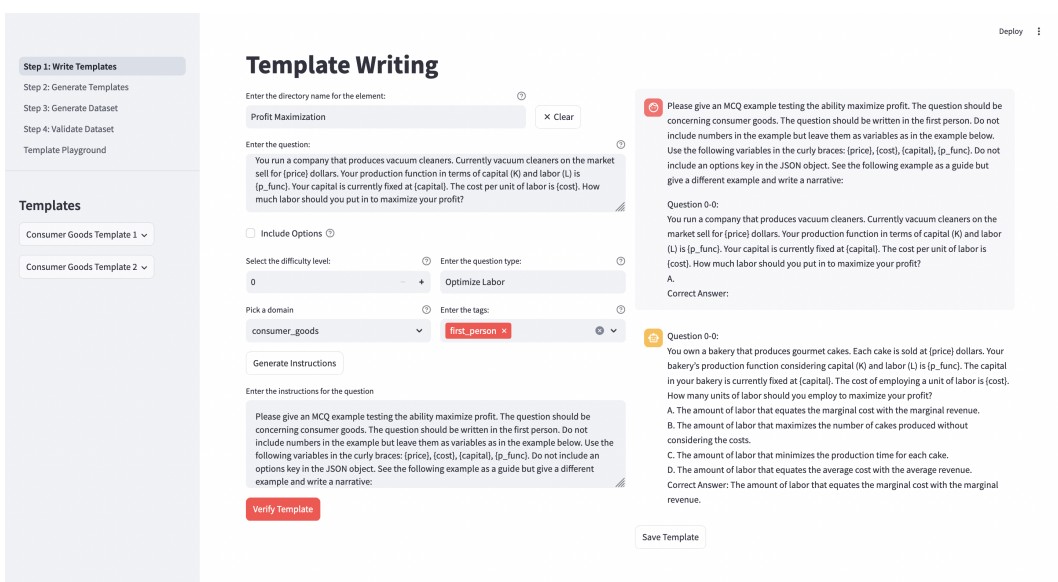

Figure 10: The web app user interface for template writing. This page includes fields for type, domain, grade level and tags (including perspectives). The right shows an example of template verification which uses a LLM to generate another template using the example seed.

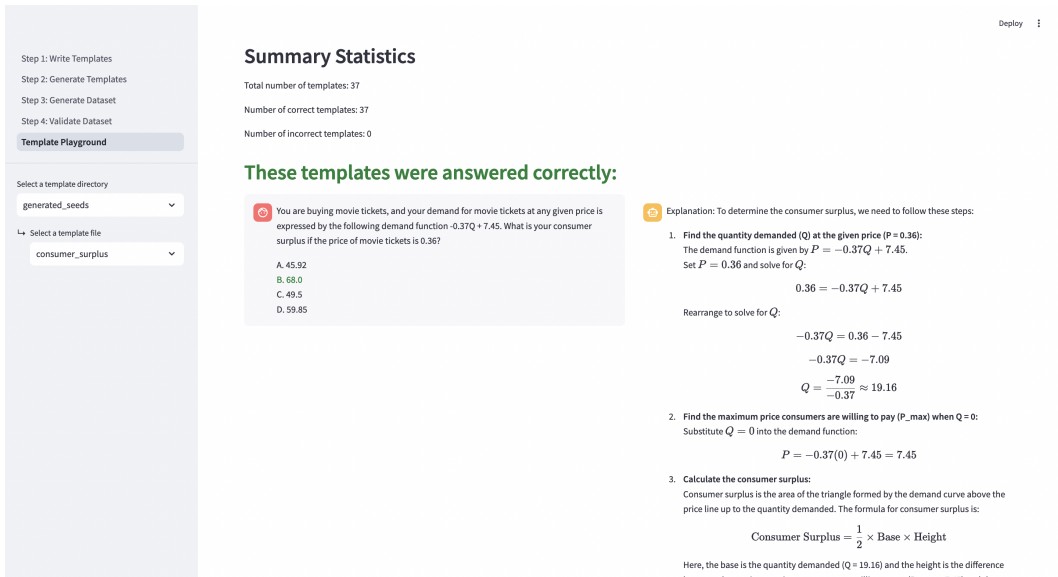

Figure 11: The web app user interface for template AI double-checking. This page instantiates and fills a set of question using a generated or example seed and then generates a response using an OpenAI model. The page also reports the number of questions answered correctly as well as the responses from the model.

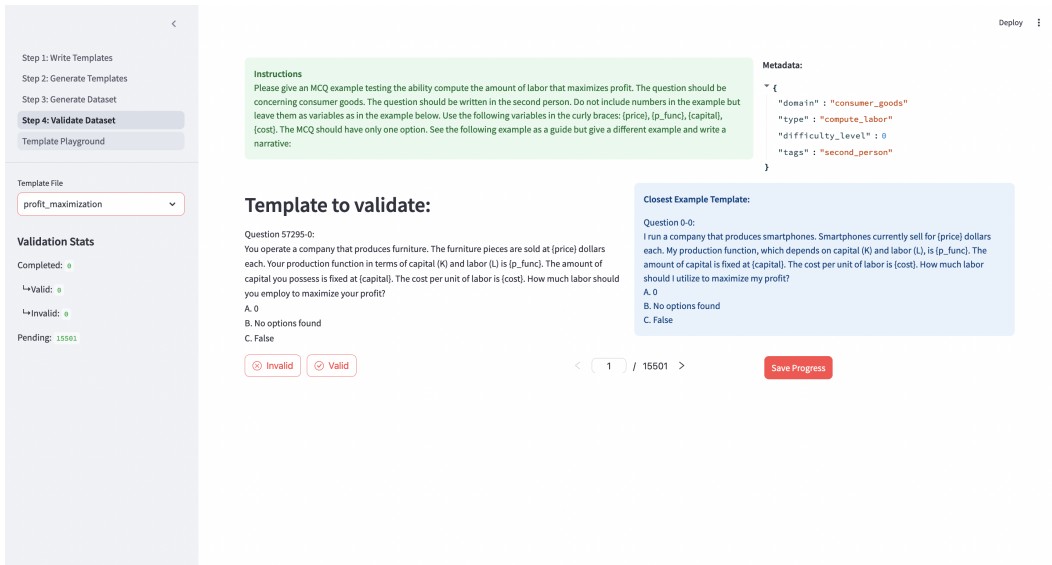

Figure 12: The web app user interface for template validation. This page displays all generated seeds returned by the model for manual validation.

