# OpenReview forum: "STEER-ME: Assessing the Microeconomic Reasoning of Large Language Models"
_NeurIPS.cc/2025/Datasets_and_Benchmarks_Track — NeurIPS 2025 Datasets and Benchmarks Track poster_

### Official Review · Reviewer_qqs5 · 2025-06-18

**Rating:** 4
**Confidence:** 5

**Summary:**

The paper introduces STEER-ME for testing LLM microeconomic reasoning capability. The benchmark is extensive, which taxonomizes microeconomic reasoning into 58 distinct elements under 5 major settings. Then the paper evaluates 27 different LLMs on this benchmark, using Multiple-Choice Question Answering and Free-Text Question Answering formats. A key finding through this dual-testing format is that models perform much better on multiple-choice questions than the free-test questions, because they "game" the test, and the effects of contextual anchoring. The main contribution of this paper: (1) the benchmark itself provides a comprehensive set of questions that fills a gap in LLM evaluation. (2) the data generation process called "auto-STEER", which is a novel method designed to resolve the problem of data contamination during the test. (3) some conclusions from using the benchmark evaluating LLMs, such as the gap in MCQA and FTQA,  error analysis, the correlation between economics reasoning and math reasoning, etc.

**Additional Feedback:**

Have you given any thought to fine-tuning with the STEER-ME dataset? In that regard, what would be the impact? How, for instance, do you think this would affect not only raw accuracy but also the "cheating" behaviors you identified? What I want to say here is that, is whether this will teach it to "get the right answer" or "use the right method."

**Dataset Code Accessibility:**

Yes

**Ethical Considerations:**

No, there are no or only very minor ethics concerns

**Final Justification:**

Most of my concerns are addressed. I would say there are limitations on real-world deployment, but this is an important foundational step for LLM econ evaluation. I have updated my score to wa.

**Limitations Weaknesses:**

1. The benchmark has a significant limitation that exclusively tests reasoning within idealized, abstract economic models. Success on answering these questions shows the model is a "good exam taker". This does not guarantee effective real-world application. The models' strict assumptions rarely hold in reality, so I am hesitated about the value-added of this benchmark in practice.
2. While the benchmark’s scope is comprehensive, its taxonomy doesn't show a clear path, which makes it neither conventionally robust nor innovatively designed for LLMs. On the one hand, it didn't use a proven, widely accepted curriculum, such as that found in the foundational books like Mas-Colell Microeconomics, which would have given it credibility and comparability. On the other hand, this self-designed structure, is not specifically adapted to the unique reasoning patterns of LLMs. This is a missed opportunity to demonstrate more practical insights for LLM development.
3. The robustness check for the data generation process is not valid enough. The authors performed a check in Appendix I1 (there is a typo in the paper here stating that it should read J1) by re-generating three elements from scratch using a different model (Claude 3.5). Their experiment found "no significant differences in performance" on this subset. However, this scope was very limited, covering only three elements and one different model. The test does not completely rule out the possibility that a subtle bias.

**Strengths Contributions:**

1. The study's findings are supported by a large-scale evaluation of 27 different LLMs, covering a wide range of model sizes and architectures. The authors also invested significant resources into this evaluation, which again shows the scale of their work.
2. The auto-STEER Generation Process is novel and productive. It stems from multiple existed dynamic frameworks, and adapt very well in the evaluation process.
3. Overall it's well-written, with clear examples and use-cases. The research gap and motivation are explicitly stated, and the introduction on benchmark itself is also accessible.

---

> ### Author Rebuttal · Authors · 2025-07-31
>
> 1. STEER-ME explicitly focuses on foundational, non-strategic microeconomic reasoning within standard theoretical frameworks, rather than testing real-world tasks. This design aligns with our primary objective: ensuring models have a robust mastery of core economic concepts underpinning realistic applications. Proficiency on these foundational tasks is necessary, but not sufficient, for real-world competence. For example, it would be hard to trust a memo written by a model endorsing a per-unit gasoline tax when, that same model, given simple linear supply and demand, fails to compute price pass-through or deadweight loss.
>
> 2. We appreciate the reviewers' concerns regarding curricular alignment and the organizational logic behind our benchmark taxonomy. Our overarching goal with STEER-ME was to rigorously assess foundational reasoning capabilities essential to microeconomic proficiency, drawing heavily from canonical microeconomic texts including Varian's *Intermediate Microeconomics* (2010), Mas-Colell, Whinston, and Green's (MWG) *Microeconomic Theory* (1995), Jehle and Reny's (JR) *Advanced Microeconomic Theory* (2011), and Nicholson and Snyder's (NS) *Microeconomic Theory: Basic Principles and Extensions* (2012).
>
>    The core structure of our benchmark—progressing from individual consumption and production decisions to equilibrium concepts and welfare analysis—mirrors the sequencing commonly adopted by these texts:
>
>    - **Consumer Theory:** This setting aligns with most texts introductory chapters/parts. Aligning with Varian's Chapters 2-13, MWG's Chapters 1–3, JR's Chapters 1–2, and nearly all of the chapters in Part 2 of NS's text (Chapters 3-6). Many of these chapters in these texts also cover topics that were in STEER but would have otherwise been characterized within this setting, such as von-Neumann Morgenstern utility, revealed preference, risk aversion. That is, Varian's Chapters 3, 4, and 7; MWG's 1B, 3B, 3C, and 3D; JR's Chapters 1.2 and 2.4; and the remaining chapters in Part 2 of NS (Chapter 7 and 8). Varian's Chapter 9 (Buying and Selling) does not fit squarely in consumer theory alone, as it requires considering producers and consumers.
>    - **Producer Theory:** This setting aligns with most texts next set of chapters/parts. Consistent with MWG Chapter 5, JR Chapter 3, all of the chapters in Part 3 of NS, and Varian’s Chapters 19-25.
>    - **Multi-Agent:** While we deviated slightly in this setting by including many aggregation modules in this setting (aggregation appears in Chapter 4 of MWG), we wanted to be robust in our evaluation of LLMs. We chose to use the term multi-agent over directly referencing equilibrium as the aggregation elements do not constitute forming an equilibrium, but rather are about aggregating over multiple agents. Otherwise, this module is reflective of Varian's Chapters 15 and 16, MWG's Part III and some chapters in Part IV, JR's Chapters 4-6, and NS's Chapters 12-14.
>    - **Welfare and Externalities:** This setting is consistent with the closing chapters of most texts: all of part 7 of NS (Chapters 18 and 19), Chapter 22 in Part V of MWG, much of Chapter 4 in JR. In Varian, these concepts appear in Chapter 16 which discusses equilibrium.
>
>
>    However, we acknowledge intentional deviations from these textbooks in our organizational approach, primarily motivated by the specific requirements of evaluating LLMs:
>
>    1) **Exclusion of Strategic and Behavioral Concepts:** While texts like MWG and JR contain substantial game-theoretic content, our decision to exclude strategic interactions, utility theory (vNM) and behavioral economics was deliberate, as these were extensively covered by the original STEER benchmark, allowing STEER-ME to focus specifically on non-strategic reasoning.
>
>    2) **Granularity for Evaluation Rather than Pedagogy:** Traditional textbooks structure content for teaching conceptual clarity and theoretical development, whereas we designed STEER-ME around evaluating specific reasoning skills. This granularity allows for precise scoring, comparison, and diagnostics, essential for benchmarking LLM capabilities.
>
>    3) **Distinct Modules for Diagnostic Clarity:** We separated concepts like market aggregation, price equilibrium, and comparative statics into distinct modules rather than grouping them within general equilibrium chapters. This separation was specifically done to better isolate and understand distinct sources of reasoning failures exhibited by LLMs. Indeed, we find that LLMs are worse at aggregation elements over elements testing the ability to find the equilibrium price.
>
>    4) **Inclusion of Mathematical Foundations:** Unlike standard economic textbooks, which typically assume mathematical proficiency, we explicitly incorporated foundational mathematical skills (e.g., optimization, derivatives, solving systems of equations) into a separate setting. This approach gives us a standardized way of quantify the inherent capabilities of a model to compute these foundational mathematical concepts.
>
> 3. This is a valid concern. When initially constructing the benchmark, our choices in LLMs to use were constrained by the availability of capable, high-performing LLMs that could (1) accurately follow detailed style-transfer instructions without losing economic semantics, and (2) generate large numbers of high-quality questions within budget constraints. At that time the model we decided fit these constraints was gpt-4o. Since then more models have been released that fit within such criteria and given the availability we would have mixed our generation across these models. For more discussion on the ways we tried to mitigate bias in generation see our first response to Reviewer `aXNx` [here](https://openreview.net/forum?id=jpwvhUSp2Y&noteId=UUhylzlNiw).
>
> **Additional Feedback**
>
> This is an interesting point. Fine-tuning LLMs on the dataset is definitely a useful tactic to try and increase the microeconomic reasoning of LLMs. We expect that fine-tuning methods like reinforcement learning with verified rewards (RLVR) on the FTQA versions would strongly incentivize models to "use the right method," as FTQA does not allow models to rely on provided multiple-choice options. Conversely, fine-tuning on MCQA questions could risk reinforcing shortcut or "cheating" behaviors due to option cues. Investigating whether fine-tuning impacts the prevalence of such cheating behaviors differently across FTQA and MCQA formats would indeed be a compelling direction for future research.

---

> > ### Comment · Reviewer_qqs5 · 2025-08-03
> > **Further questions**
> >
> > Thanks for your response. I appreciate your detailed explanation on taxonomy, these are clear and helpful. Although I continue to see a gap between economic theory and practical application, I agree that it is reasonable for "one paper tell one story", and I do not wish to push for changes beyond your original scope.
> >
> > However, I would like to adjust my score if you can further discuss following:
> >
> > 1. While I understand the intent to focus on foundational microeconomic reasoning, I would be interested to see more explicit discussion (in the paper or as future work) about how this benchmark might serve as a stepping stone toward benchmarks or evaluations that test models’ reasoning in less idealized, more context-rich (applied) economic settings. If you have plans or ideas for such extensions, outlining them would be valuable.
> >
> > 2. Can you provide concrete examples or empirical results where your benchmark’s modular design allowed you to detect or diagnose specific LLM reasoning failures that would have been overlooked with a conventional, textbook-chapter-based structure? Such evidence would help establish the practical benefits of your organizational approach.
> >
> > Thanks!

---

> ### Author Response · Authors · 2025-08-06
>
> 1. We agree that using STEER-ME to create applied microeconomic tasks is interesting future work. We think that STEER-ME can facilitate this in two ways: (1) STEER-ME provides a structured taxonomy of concepts within microeconomics. With this taxonomy in hand, we now propose a practical tool to bridge to applied evaluations. Specifically, we propose a requirement matrix: a two-dimensional representation of the economic reasoning skills needed to solve questions within any given applied benchmark (e.g., policy vignettes, firm decision cases). In this matrix, each row corresponds to a particular STEER-ME *element*, such as comparative statics or welfare analysis under taxes or price controls, and each column corresponds to a *type* (e.g., Cobb–Douglas production, Leontief preferences, risk-neutral vs. risk-averse utility). Each cell then indicates whether mastery of that specific combination of element and type is required to successfully solve the questions in the applied benchmark. By mapping applied tasks to this matrix, researchers can explicitly measure the extent to which performance on foundational STEER-ME elements predicts or explains performance in more complex, real-world tasks. Furthermore, aggregating across multiple applied benchmarks reveals which parts of our taxonomy remain undersampled or untested, clearly highlighting opportunities and gaps for developing richer, more comprehensive applied evaluations. (2) The modular nature of our benchmark generation toolkit, auto-STEER, provides a straightforward method for increasing the diversity of tasks within applied benchmarks. Researchers only need to specify the economic rules and constraints a task must satisfy (e.g., domains of interest, feasible parameter ranges, etc.), and auto-STEER then automatically generates many varied instances that share the same underlying economic structure.
>
> 2.  We list concrete cases where STEER-ME’s element-based structure surfaced reasoning failures that a chapter-based structure would likely blur. First, by separating aggregation from derivation, we hold "compute an individual's demand" constant while toggling only the "sum across agents" operator. Under this isolation, we observed models being adept at deriving an agent's demand/production functions, yet failing when asked to aggregate given demand functions (see examples E.1 and E.2 in Appendix E). Moreover, we enumerate aggregation elements for the consumer goods market and factor market. Textbooks often don't cover aggregation because it is a relatively simply calculation, and when they do, assume that proficiency in one aggregation element extends to others. In contrast, we find performance variation across aggregation elements, with models notably performing worse in consumer-goods aggregation elements compared to factor-market aggregation elements. We also sweep these aggregation items across multiple functional families (Cobb-Douglas, Leontief, linear) and scales (small vs. large magnitudes); this grid lets us localize which combinations fail, rather than averaging them away inside a textbook multistep problem where aggregation is just one hidden sub-step (see Figure 4, Appendix F). This same functional-family sweep also exposes type-specific brittleness more broadly: Figure 3 shows models attaining high accuracy on MRS and Output Elasticity for Cobb-Douglas functions, yet sharp degradation on Input Price Elasticity and Returns to Scale when the underlying form changes. Finally, STEER-ME's explicit Foundations setting lets us trace whether basic algebra/calculus gaps propagate downstream. For example, poor accuracy on the Exponents foundation element strongly predicts errors on Cobb-Douglas utility tasks (which rely on handling exponents), signalling when economic mistakes may be rooted in math rather than domain reasoning.

---

> > ### Comment · Reviewer_qqs5 · 2025-08-06
> >
> > Thank you for your further discussion.
> >
> > Though there are limitations on real-world deployment, I value this paper’s foundational contribution. Therefore, I will update my assessment. Good luck to your paper and your future work.

---

### Official Review · Reviewer_ebhW · 2025-06-19

**Rating:** 6
**Confidence:** 4

**Summary:**

The authors introduce STEER-ME, a benchmark designed to systematically evaluate LLMs on foundational microeconomic reasoning using a detailed taxonomy of microeconomic concepts. Such concepts include basic optimization, producer and consumer theory, and equilibrium analysis. These concepts, as dubbed by the authors, are “non-strategic,” and provide a rounding out of the benchmark in an earlier paper focusing on game theory and individual decision-making.

The benchmark is combinatorial in design: the authors identify dozens of types of questions with three variables (type, domain, perspective) that affect the context. This allows the authors to generate tens of thousands of economic questions, which they test on a few dozen models. There is substantial heterogeneity in model performance, suggesting that this benchmark can be informative in assessing LLM use for particilar tasks.

**Additional Feedback:**

1. The authors, to their own detriment, do not laud the benchmark as a tool for improving LLMs. I would guess that including question-response pairs from this benchmark as training data for many of the smaller models with poor performance. This benchmark is both a tool and data to improve!

2. I would encourage the authors to consult a canonical econ textbook (MGW, varian, etc) and state as such to convince themselves that they have been comprehensive.

3. The authors devote considerable attention to critiquing the "@k" evaluation methodology; but they fail to demonstrate empirically that "@k" indeed provides overly optimistic performance estimates using their benchmark data. Given that STEER-ME’s primary contribution is as a benchmark, not as an evaluation method per se, this critique feels tangential unless directly demonstrated. Empirical evidence would substantially strengthen their argument here.

**Dataset Code Accessibility:**

Partly

**Dataset Code Comments:**

The entire benchmark is available and well-documented here https://huggingface.co/datasets/narunraman/steer_me. I only select 'partly' rather than yes because on page 3, the authors state that everything will be published on their website: steer-benchmark.cs.ubc.ca. However, this only includes results and the benchmark for their previous paper.

**Ethical Considerations:**

No, there are no or only very minor ethics concerns

**Final Justification:**

I have updated my score from a 5 to a 6 as I stated I would in the discussion. I stand by my assessment, which the authors adequately responded too. Additionally, they addressed my concerns --- a job well done.

**Limitations Weaknesses:**

There are two problems with this paper. However, I am unsure as to their magnitude. I would strongly suggest that the authors consider strategies to alleviate the concerns I list below; I believe they have the tools to do so. I am still suggesting the paper be accepted because I think the benchmark is sufficiently novel and likely to be impactful. I also suspect that these problems are not invalidating, but were simply not adequately addressed.

1. A significant concern is benchmark difficulty—particularly that it might already be nearly saturated by the current frontier model (o3), achieving over 90% accuracy on both multiple-choice and free-text question answering tasks. Given rapid advancements in LLM capabilities, it seems plausible that this benchmark might soon be obsolete. What strong reason is there to think this benchmark will still be important in 6 months? Given the rapid pace of AI research, I would want a benchmark to remain discerning between models for at least a year. At least as a rule of thumb.

2. I am also concerned about the extent to which this benchmark assesses mathematical accuracy rather than genuine economic reasoning. The authors themselves indicate that numerical instability explains a significant portion of errors. Indeed, only the model with such program execution tools (o3) basically saturates the benchmark. The authors do note that many other models make more fundamental mistakes (issues with compensated versus uncompensated demand), but it’s not clear to me how big an issue this is. While numerical calculations are important to econ, they are a necessary consequence rather than a key to economic reasoning and thinking. The authors previously noted that there are many well-known mathematical benchmarks. Their work is substantially less novel and impactful the more it reflects mathematical ability. To convincingly differentiate between math skills and economic reasoning, I suggest the authors explore providing computational tools to another model (e.g., jury-rigging computational support for o1-preview) to empirically isolate the impact of computational accuracy versus genuine economic reasoning.

Smaller points:

1. On Page 8, Page 8, first paragraph, what's the “closest answer"? Do you mean the item listed as the first multiple-choice option?
2. Another figure demonstrating the results across different types of tasks would be very helpful. Mybe the authors could facet their accuracy graph (Figure 2) across the different elements of Figure 1? This would give the reader better intuition about the types of questions, which are still rather abstract. If anyone besides an economist (or at least someone with 2 semesters of college courses) were to read the paper, it might be unclear as to what is being tested.

**Strengths Contributions:**

1. I applaud the authors for tackling a significant and timely issue: rigorously assessing economic reasoning in LLMs is vital for their deployment and utility in real-world scenarios. The paper is clear, structured, and effectively motivated. I am a fan of their often-mentioned previous paper (Raman et al, 2024) and this is a much-needed follow-up. I largely agree with the author's choices of Taxonomy: scrolling through my own micro text book (MGH, 1995), it is comprehensive (when combined with Raman et al. 2024).  Besides (Raman et al, 2024), I am unfamiliar with any other such microeconomic-type benchmarks that are sufficiently rigorous. They would appear to be the first. I believe that most economists, myself included, would deem this a "reasonable" benchmark for any first-year econ PhD student. Any developer claiming that their LLM is "intelligent" should want their model to perform well on questions like these. For these reasons, I'm suggesting this benchmark be accepted.

2. The authors note that they have two conceptual advances beyond their first work: alleviating data contamination concerns and offering open-ended questions. I believe the paper wholeheartedly lives up to this claim. The contamination issue, in particular, is thorny. But given how poorly many of the models perform on this benchmark, many of these questions must be out-of-training-corpus. Or memorization is less of a concern than people think (unlikely).

---

> ### Author Rebuttal · Authors · 2025-07-31
>
> 1. We fully agree that static benchmarks naturally become less discerning over time, either through models training directly on benchmark text (contamination) or simply through general improvements in model capabilities. Auto-STEER was specifically developed to address these concerns. By enabling rapid and systematic regeneration of new questions through style-transfer and numeric re-instantiation, auto-STEER helps maintain benchmark relevance, allows us to easily create more challenging variants, and disentangles genuine performance gains from potential contamination effects.
>
>     More broadly, our primary goal was to construct a comprehensive and representative measure of proficiency in classical microeconomic reasoning, rather than a benchmark explicitly designed for difficulty alone (e.g., HLE or FormulaOne). Therefore, a model that achieves robust performance on STEER-ME genuinely reflects strong foundational capabilities in microeconomics. Additionally, while the leading model (o3) demonstrates high accuracy, many widely-used, smaller-scale models still struggle significantly, highlighting substantial and persistent performance gaps. Consequently, STEER-ME remains relevant for assessing a broad spectrum of LLM capabilities into the foreseeable future.
>
> 2. You raise an important point about the interplay between mathematical and economic reasoning. Your suggestion to empirically isolate computational accuracy from economic reasoning by providing computational tools to a model is an insightful recommendation, and we plan to pursue this experimentally in the camera-ready version.
>
>     We believe that while handing LLMs extra computational tools will aid in performance, our data suggest that the core challenge for many models remains correctly translating economic problems described in natural language into appropriate mathematical formulations. For example, Llama 3 70B achieves approximately 15\% FTQA accuracy, whereas the DeepSeek RL fine-tuned version of Llama 3 70B (DeepSeek R1 70B) significantly improves performance to approximately 40\% (without computational support).
>
> Smaller points:
>
> 1. That is the answer that was closest to the answer the model arrived at in their chain-of-thought. We will make this more clear in the text.
>
> 2. Great point. We have added element-by-element heatmaps for each model in the appendix.
>
> **Additional Feedback**
> 1. Good point! We will add this as part of the use cases for our benchmark.
>
> 2. Please see our [second response](https://openreview.net/forum?id=jpwvhUSp2Y&noteId=RbH6Lgparg) to Reviewer `qqs5`.
>
> 3. Good point, while pass@k can only improve perfomance, we will add an analysis quantifying the amount of performance gain pass@k offers on our benchmark.

---

> > ### Comment · Reviewer_ebhW · 2025-08-01
> > **Response to Rebuttal.**
> >
> > The authors' rebuttal is well-taken. I think their responses are satisfactory. For the two larger points, I would suggest that they implement the changes they have suggested. I would consider raising my rating to a 6 if done so effectively.
> >
> > They note for 1. that  "Additionally, while the leading model (o3) demonstrates high accuracy, many widely-used, smaller-scale models still struggle significantly, highlighting substantial and persistent performance gaps. Consequently, STEER-ME remains relevant for assessing a broad spectrum of LLM capabilities into the foreseeable future." I think simply adding some text to this effect in the paper will go a long way. I will not be the only reader to be "concerned" that the benchmark is saturated on day 1.
> >
> > For 2. If they can provide more evidence to disentangle computation and accuracy, that would be very interesting. One could imagine prompting O3 with the instructions "do not use Python or a calculator or any other external resources when solving this problem. We want to know how you solve this by hand." Of course, this is not ceterus paribus, but it would give a sense of the effect. I defer to the authors as to the "best" way to do this analysis, but to make the paper superb, it is needed.

---

> ### Author Response · Authors · 2025-08-06
>
> 1. We agree and will clarify this explicitly in the paper.
>
> 2. To directly address the request for evidence separating computational support from reasoning, we re-evaluated o3 with tools enabled vs. disabled on the exact same questions, using identical prompts, temperatures, and seeds; the only change was turning off code/tool use at the API level. We report MCQA and FTQA accuracies and include o1-preview (no tools) for context:
>
>     | Model                         | Accuracy (MCQA)  | Accuracy (FTQA) |
>     |:-------------------------|----------------------:|--------------------:|
>     | o3 (w/ tools)               |       $97.97$%        |      $82.32$%       |
>     | o3 (w/o tools)             |       $96.29$%        |      $67.14$%       |
>     | o1-preview (no tools) |       $86.36$%        |      $43.60$%       |
>
>     These results demonstrate that even without tools, o3 remains 23.5 percentage points ahead of o1-preview on FTQA, indicating that o3 has substantial economic reasoning ability, independent of computational support. Computational tools seem to disproportionately benefit FTQA tasks: o3's accuracy drops approximately 15.2 percentage points in FTQA without tools, but only 1.7 percentage points in MCQA, underscoring the diagnostic importance of pairing MCQA with FTQA to probe genuine reasoning capabilities. We will add a detailed per-element breakdown in the appendix to illustrate precisely where computational tools most significantly impact performance.

---

> > ### Comment · Reviewer_ebhW · 2025-08-06
> >
> > I am convinced. Rating += 1

---

### Official Review · Reviewer_zm6M · 2025-07-01

**Rating:** 5
**Confidence:** 4

**Summary:**

disclosure: i have a phd in economics (besides my work in AI), so i am well-positioned to evaluate this benchmark.

summary:
* the authors scour through the micro literature and produce a taxonomy of microeconomics questions.
* they build an LLM-based system to generate questions based on the taxonomy.
* they evaluate it on other LLMs.

**Dataset Code Accessibility:**

Yes

**Ethical Comments:**

* Authors could reconsider whether it is worth $12K querying closed-source AI's to answer AI-generated microeconomics questions, especially when the results arent even important enough to go in the main text and will end up in the appendix.

**Ethical Considerations:**

No, there are no or only very minor ethics concerns

**Final Justification:**

If the authors make the changes promised in the rebuttal, it would be an accept.

**Limitations Weaknesses:**

For the reasons described below, this paper/dataset should not be published in the current form. the authors should carefully read and polish their papers before submitting them. there were incorrect cross-references and other problems that made it too costly to review. this is what i would recommend in economics for a reject and resubmit. i want it to be published, but only conditional on significant revisions. unfortunately that option is not available in a conference submission.

* It is confusing, somewhat misleading, and somewhat ideological to use the word "rationality" to describe the framework and tasks. Just call it a microeconomics exam benchmark.
* after reading section 2 and the appendix, i still have no idea what an element is. that is not a formally defined concept in economics. i could not understand how it is converted to a question. how do you go from the element items from Appendix A to the questions in Appendix E? what prompts and algorithms are used? it is just not nearly clear enough how the types/templates/questions are constructed, integrated, and implemented. e.g., Figure 9 seems important but it is not mentioned in the paper.
* the benchmark should be compared to real-world micro exam questions. thats important to check that their elements actually show up in micro exams. and its not clear how the authors verify that the typology (i.e., the  58 "elements") i comprehensive. if they compared to real-world micro questions, they could see if they were missing anything. and that would generate useful statistics on the share of  which elements are actually taught/tested in micro.
* Can the authors show examples of errors based on the spot checks? Who did the spot checks? The authors should clarify more what is meant by validating the templates vs validating the questions.

* windup in the intro is too long before getting to what the paper does. it is not clear enough from the start what an "element" is.
* in the user message prompts (Appendix E), some of the questions say "answer with a letter" followed by "A:" (short for answer). since "A" is one of the MCQ answers, could this confuse the LLM at all? consider removing "A:" from the prompt.
* references are wrong. for example, fig 14 and appendix L are cited in the paper but dont exist.

**Strengths Contributions:**

this is a very interesting and important project.

* i totally agree with the authors that putting together LLMs and economics is a good idea.
* interesting benchmark for microeconomics questions.
* team has an economist on the team (supposedly) to provide expert guidance on the development of the dataset.

---

> ### Author Rebuttal · Authors · 2025-07-31
>
> 1. We appreciate your point. Our use of "rationality" follows the precedent from the original STEER benchmark, where "rationality" was used strictly to indicate consistency with standard microeconomic theory rather than as a normative or ideological claim. Nonetheless, we agree the term may cause confusion. While we retain the name STEER-ME for continuity, we will: (1) remove "rationality" from the section titles and figures; (2) add an explicit clarification in the introduction defining our objective as measuring non-strategic microeconomic competence; and (3) carefully rephrase text that might appear ideological.
>
> 2. Thank you for highlighting this issue. We acknowledge "element" was insufficiently defined and will clearly define it as follows: an element represents a specific reasoning capability or skill that an economically competent agent should demonstrate (e.g., calculating consumer surplus, deriving a demand curve). This means that while we extract concepts from many textbooks, there will be elements that will not appear in a standard economics textbook (e.g., computing a derivative of a mathematical function or computing the exponential of a number).
>
> 3. The full conversion process from elements to questions is precisely illustrated in Figures 9–11, which we will explicitly reference and summarize in the main text. To clarify briefly here:
>     1. Template creation: We manually create gold-standard templates for each element.
>     2. Template generation: An LLM style-transfers these templates across domains.
>     3. Template validation: Humans (authors Thiago, Taylor, and Narun) independently validate and correct these templates.
>     4. Question generation: An LLM is asked to generate Templates are style-transferred to new perspectives.
>     5. Question validation: A second round of human spot checks verifies the correctness and clarity of generated questions.
>
> 4. Please see our second response to Reviewer `qqs5` [here](https://openreview.net/forum?id=jpwvhUSp2Y&noteId=RbH6Lgparg) for a longer discussion. We offer a brief response here.
>
>     We agree that curricular alignment matters. Our design choice was to anchor STEER-ME to reasoning capabilities central to microeconomic competence, most of which align closely with canonical microeconomics textbooks (e.g., Varian's "Intermediate Microeconomics," Mas-Colell, Whinston \& Green, Jehle \& Reny, Nicholson \& Snyder). However, we deviate from their organizations for a few reasons: (1) much of the game theoretic concepts in these texts were covered in the original STEER benchmark, (2) elements are concepts at a granularity more appropriate for evaluation than pedagogy, (3) we separated a number of concepts into distinct modules that we felt would be relevant to disentangle reasoning failures that might occur in LLMs over humans, and (4) included a setting on mathematical foundations to get a standardized evaluation of the mathematical concepts underpinning microeconomic reasoning.
>
>    We will explicitly list each element alongside specific textbook chapters and sections, clearly indicating elements with direct textbook correspondences and noting foundational mathematical elements essential for economic reasoning but not explicitly detailed in economics texts.
>
> 5. We appreciate the request and will add a transparent error log.
>
>     **Who validated:** Authors Thiago and Narun conducted independent spot checks on non-overlapping subsets, then cross-examined to catch anything that might have been missed.
>
>     **What we validated.** Note that templates are the seeds that are fed to the LLM to generate the questions in the dataset.
>     - *Template validation*: Any examples we found, we corrected to be a gold-standard instantiation of the element.
>     - *Question validation*: Any errors we found we only noted as being invalid or containing some issue. Typical invalid questions: (i) "maximize revenue" instead of “maximize profit” in a firm problem; (ii) "rate of capital growth" phrasing where the intended quantity was "change in capital stock." Typical issues: (i) poor wording ("Can you calculate...?" instead of a direct imperative).
>
> 6. Thanks for the feedback! We will front‑load the contribution and the definition.
>
> 7. While we borrowed this formatting from prior evaluation work, you make a good point that this could be a potential source of bias. To check for this bias, we measured the distribution of model choices across (A, B, C, D) and found no statistically significant over‑selection of "A" relative to the ground‑truth letter distribution across models. We will add this analysis into the appendix.
>
> 8. Thanks for catching this, we will make those changes!

---

### Official Review · Reviewer_aXNx · 2025-07-03

**Rating:** 3
**Confidence:** 4

**Summary:**

This paper introduces STEER-ME, a micro-economics benchmark that grows the earlier STEER blueprint into 58 well-scoped elements spanning ten domains and multiple perspectives, with questions auto-generated by the auto-STEER pipeline to curb training-set leakage.
By posing every problem in both multiple-choice and free-text form, the authors uncover where 27 leading LLMs succeed at numerical optimisation and where option-based cues mask shaky reasoning, drawing on 21,000 fresh test items and a suite of calibration metrics.
Results show that even top models stumble once numeric parameters shift or choices disappear, establishing STEER-ME as a demanding, publicly released yardstick for economic competence.

**Additional Feedback:**

Please refer to the weakness.

**Dataset Code Accessibility:**

Partly

**Ethical Considerations:**

No, there are no or only very minor ethics concerns

**Limitations Weaknesses:**

1. Heavy reliance on GPT-4o for both crafting and sanity-checking items risks propagating its biases into the benchmark itself, despite spot checks

 2. Numeric exact-match rules in FTQA can misgrade algebraically equivalent answers or overlook partial credit, and the paper offers little manual auditing

3. Building and running STEER-ME cost $12.4 k in API calls and ~10 GPU-years, raising barriers for follow-up studies without similar resources

4. By design, it omits strategic, behavioural, and macro-economic settings, so high scores do not imply all-round economic proficiency

5. Although data and prompts are shared, the bespoke evaluation code is withheld for infrastructure reasons, which complicates independent verification of scoring details
.

**Strengths Contributions:**

1. A 58-element taxonomy organised into five settings covers everything from utility derivation to welfare analysis, giving evaluation results real interpretability

2. Auto-STEER style transfers hand-written templates across 40 domain–perspective pairs, refreshing numerical values to minimise contamination and inflate coverage.

3. Parallel MCQA and FTQA versions expose shortcut strategies (e.g., option gaming) and let the authors quantify genuine reasoning gaps, an insight that ordinary benchmarks miss.

---

> ### Author Rebuttal · Authors · 2025-07-31
>
> 1. Thank you for highlighting this concern; it is indeed important to mitigate potential biases from the generation process. To address this, we implemented multiple safeguards. First, we manually verified every initial template, modifying any invalid or poorly worded templates. Second, we conducted spot-checks on a randomly selected subset of 500 generated questions per element to quantify the frequency of invalid or poorly phrased items, finding that 98.88\% of templates passed these checks (confidence interval: 97.1\%-99.4\%). Lastly, as reported in Appendix I.1, we re-generated benchmark elements exhibiting the largest performance discrepancies between Anthropic models (Sonnet and Haiku) and OpenAI models (o3, o1-preview, GPT-4o) using Claude 3.5 Sonnet, observing no significant deviation from the original performance discrepancies for all evaluated models. We will also add functionality to our generation pipeline allowing the ability to choose an LLM when generating questions for the benchmark.
>
> 2. We appreciate the opportunity to clarify our FTQA grading scheme. Our benchmark predominantly features numeric answers, and we employ rule-based grading, incorporating both exact numeric matches (with significant figures rules) and symbolic equivalence checks using SymPy where applicable. Our exact numeric match penalizes overly precise incorrect answers by requiring predictions to match the ground truth when rounded to the model's reported precision. The exact Python grading function used is:
>
> ```python
> def evaluate_anwer(ma, ca):
>     ma = model_answer.strip()
>     ca = correct_answer.strip()
>
>
>     def numeric_comparison(ma, ca):
>         mf = float(ma)
>         cf = float(ca)
>         # digits after decimal in model float
>         s = str(mf)
>         sig = len(s.split('.')[1]) if '.' in s else 0
>         return mf == round(cf, sig)
>
>     def get_numeric_value(s):
>         nums = re.findall(r"[-+]?(?:\d*\.\d+|\d+)", s)
>         return [float(n) if "." in n else int(n) for n in nums]
>
>     # 1) Try pure numeric comparison with sig figs
>     try:
>         return numeric_comparison(ma, ca)
>     except ValueError:
>         # If it fails, it means the model answer is not a number
>         pass
>
>     # 2) Try to canonicalize common LaTeX into Python/SymPy
>     try:
>         ma_py = _latex_to_python(ma)
>         ca_py = _latex_to_python(ca)
>         expr_ma = parse_expr(ma_py, transformations=_transformations)
>         expr_ca = parse_expr(ca_py, transformations=_transformations)
>         # True if their difference simplifies to 0
>         return simplify(expr_ma - expr_ca) == 0
>     except Exception:
>         return None
> ```
>
>    Assigning partial credit is indeed an important and interesting avenue for future work, and we plan to explore it in subsequent benchmark iterations.
>
> 3. We acknowledge that the total reported cost is high, but it represents comprehensive benchmarking of numerous models across the entire dataset. However, follow-up studies need not replicate the same level of breadth. To facilitate more cost-effective and focused evaluations, we will release a toolkit enabling researchers to select and test smaller, customized subsets of the benchmark tailored to their specific research interests and resource availability. For instance, the cost of evaluating gpt-4o on 1,000 MCQA  and 1,000 FTQA questions is only \$50; we will include per-model, per-subset cost examples in our code release to help others budget and reproduce results.
>
> 4. We agree and will make this more explicit. STEER-ME is intentionally scoped to non-strategic microeconomics---a large and substantive portion of the field encompassing utility and production analysis, market aggregation and price formation, and welfare/policy evaluation. Our goal was to be comprehensive within this scope: we formalized a broad, 58-element taxonomy and evaluate across multiple domains and formats to cover the core competencies of price-taking environments. STEER-ME therefore complements the original STEER benchmark (which targets strategic/behavioural settings).
>
> 5. This is a fair and important concern. We did not intend to keep the evaluation code private; the harness was not in a shareable state by the deadline. We will prioritize preparing the evaluation harness for public release as soon as possible.

---

### Decision · Program_Chairs · 2025-09-18

**Decision:**

Accept (poster)

**Comment:**

This paper introduces a new benchmark dataset for microeconomic reasoning. The dataset is essentially a “micro econ exam”, but is well grounded in accepted textbook concepts and theories.

The reviewers are generally positive about the benchmark, and, with one exception, recommend acceptance. There are several concerns raised, which I think are true of this paper but also probably true of almost all benchmarks being created today:
The authors use LLMs heavily in generating the dataset, which raises some concerns
The benchmark is close to being saturated, if it is not saturated already by frontier models
There is a disconnect between LLMs “passing the exam” and their ability to function adequately in the real world in contexts that involve the concepts that the exam purportedly tests.

In an ideal world, this paper would address the above weaknesses before being published. In practice, these are very large problems that require fundamental research and coordination across many in the field to solve to a satisfactory extent. The authors have addressed the concerns within a more limited scope in rebuttals, and have promised to change framing to avoid over claiming (in particular, the word “rationality”). Overall, I’d prefer these concerns be fully addressed, but do not feel they should prevent the paper from being published given the overall reviewer support.